# EGFR core fucosylation, induced by hepatitis C virus, promotes TRIM40-mediated-RIG-I ubiquitination and suppresses interferon-I antiviral defenses

Qiu Pan [1], Yan Xie [1], Ying Zhang [1], Xinqi Guo [1], Jing Wang[1], Min Liu[1] ✉ & Xiao-Lian Zhang [1,2] ✉

Aberrant N-glycosylation has been implicated in viral diseases. Alpha-(1,6)-fucosyltransferase (FUT8) is the sole enzyme responsible for core fucosylation of N-glycans during glycoprotein biosynthesis. Here we find that multiple viral envelope proteins, including Hepatitis C Virus (HCV)-E2, Vesicular stomatitis virus (VSV)-G, Severe acute respiratory syndrome coronavirus 2 (SARS-CoV-2)-Spike and human immunodeficiency virus (HIV)-gp120, enhance FUT8 expression and core fucosylation. HCV-E2 manipulates host transcription factor SNAIL to induce FUT8 expression through EGFR-AKT-SNAIL activation. The aberrant increased-FUT8 expression promotes TRIM40-mediated RIG-I K48-ubiquitination and suppresses the antiviral interferon (IFN)-I response through core fucosylated-EGFR-JAK1-STAT3-RIG-I signaling. FUT8 inhibitor 2FF, N-glycosylation site-specific mutation (Q352AT) of EGFR, and tissue-targeted Fut8 silencing significantly increase antiviral IFN-I responses and suppress RNA viral replication, suggesting that core fucosylation mediated by FUT8 is critical for antiviral innate immunity. These findings reveal an immune evasion mechanism in which virus-induced FUT8 suppresses endogenous RIG-I-mediated antiviral defenses by enhancing core fucosylated EGFR-mediated activation.

Protein glycosylation is a common and important post-translational modification that plays a central role in numerous physiological and pathological processes including viral diseases, immune response[1–4]. Fucosylation is an important subtype of N-glycosylation and catalyzed by 11 fucosyltransferases (FUT1-11) to form α1,2-, α1,3-/4-, or α1,6-linked glycosyl linkages, and two special protein O-linked fucosyltransferases (poFUT1-2)[5]. FUT8 is the sole enzyme responsible for α1,6-fucosylation (core fucosylation) through the addition of α1,6-linked fucose to N-glycans[6]. Previously, we and other laboratories found that enveloped hepatitis virus B/C (HBV/HCV)[7–9] can induce FUT8 production, which is crucial for multiple physiological and pathological processes, including cell growth, adhesion, receptor activation, ADCC, tumor metastasis, and viral infection[7–14]. However, whether other viruses can induce FUT8 expression and which critical host proteins can be modified by core fucosylation remain unknown. The detailed mechanisms underlying the role of FUT8 in viral infection remain unclear.

As obligate intracellular parasites, the survival of viruses is intricately linked to their ability to rely on the protein synthesis machinery

[1]State Key Laboratory of Virology and Hubei Province Key Laboratory of Allergy and Immunology, and Department of Immunology, Wuhan University TaiKang Medical School (School of Basic Medical Sciences), Wuhan 430071, China. [2]Department of Allergy, Zhongnan Hospital of Wuhan University, Frontier Science Center for Immunology and Metabolism, Medical Research Institute, Wuhan University, Wuhan 430071, China. ✉e-mail: mliu@whu.edu.cn; zhangxiao-lian@whu.edu.cn

of their host cells and to regulate the cellular processes necessary for viral replication[15]. It has been reported that the infection of viruses from a variety of different families causes modification of the host cell glycosylation profile and activation of host cell glycosyltransferase transcription[16–20]; however, how the viruses manipulate host transcription factors to induce FUT8 expression remains to be explored.

Epidermal growth factor receptor (EGFR, also known as ErbB1 and HER1) is a highly glycosylated and phosphorylated transmembrane receptor tyrosine kinase (RTK) that regulates several essential processes, including cell proliferation, survival, differentiation during development, tissue homeostasis, tumorigenesis, and type I interferon (IFN-I) signaling[21]. Previous studies have shown that multiple enveloped viruses (including HCV, herpes simplex virus HSV-1, influenza virus IAV, and human cytomegalovirus HCMV) interact with EGFR to facilitate viral entry and replication. The binding of HCV particles and the HCV-E2 protein to human hepatocyte CD81 induces EGFR activation and internalization[22]. EGFR activation was demonstrated to be required for influenza A virus internalization through the clustering of lipid rafts[23], suggesting that EGFR internalization may be a mechanism utilized by viruses to enter cells. Previous studies have indicated that increased core fucosylation of EGFR significantly promotes EGF-mediated intracellular signaling[24,25]. However, the effect of EGFR fucosylation on viral replication remains unclear.

Multiple important human viruses, including hepatitis viruses, severe acute respiratory syndrome coronavirus 2 (SARS-CoV-2), and human immunodeficiency virus (HIV), pose serious threats to human health, and new therapeutic and prevention strategies for these viral diseases need to be explored. IFN-I plays a critical role in anti-viral responses. However, whether FUT8 is involved in IFN-I anti-viral response remains unknown.

In this study, we identify that HCV-E2, VSV-G, SARS-CoV-2-Spike and HIV-gp120 enhance FUT8 expression and core fucosylation. HCV and HCV-E2 manipulate host transcription factor SNAIL to induce FUT8 expression through EGFR-AKT-SNAIL activation. HCV-induced FUT8 further promotes the activation of core fucosylated EGFR downstream JAK1 and STAT3 cascades. The aberrant increased-FUT8 expression promotes TRIM40-mediated RIG-I K48-ubiquitination. We find that virus-upregulated FUT8 promotes RNA viral replication through suppression of RIG-I-mediated IFN-I antiviral defense. Hepatic targeted Fut8 silencing suppresses HCV RNA replication in human transgenic mice (ICR[4R+]). Similar inhibitory effects of FUT8 on VSV replication in mice are demonstrated using the Fut8 inhibitor. Our data reveal that FUT8 negatively regulates RIG-I-mediated antiviral innate immune response through EGFR core fucosylation and RIG-I degradation.

## Results

### HCV and several other enveloped viruses upregulate host cellular FUT8 expression

Currently, the most widely used infectious HCV culture system is based on JFH1 (Japanese fulminant hepatitis 1, genotype 2a)[8], which undergoes efficient replication in Huh7 cells and other cell lines[26–29]. Our recent study has shown that HCV promotes FUT8 expression in Huh7.5.1 cells[8]. In this study, we further examined and confirmed that HCV promoted FUT8 expression in Huh7 cells, besides Huh7.5.1 cells (Fig. 1a). The expression of a panel of FUT family genes was screened using real-time reverse transcription-quantitative PCR (RT-qPCR) in Huh7 cells infected with HCV. Among these genes, the mRNA level of FUT8 was significantly upregulated 3- and 10-fold at 6 and 12 h post-infection of HCV, respectively (Fig. 1a). Aleuria aurantia lectin (AAL, derived from *Aleuria aurantia*) and Aspergillus oryzae lectin (AOL, from pathogenic fungus *Aspergillus oryzae*) have been often used as carbohydrate probes for core fucose in glycoproteins[12,30–35]. Increased

core fucosylation was detected by both AAL and AOL blotting of Huh7 cells infected with HCV (Fig. 1b).

Previous study has reported the HCV-infected ICR[2R+] (transgenic mice in ICR background harboring both human CD81 and occludin genes) mouse model[36]. Here we utilized a humanized HCV infection mouse model, which harbored human scavenger receptor B1 (SR-B1), CD81, claudin-1 (CLDN1), and occluding (OCLN) (essential receptors or coreceptors for HCV cell entry) genes (Supplementary Fig. 1a), named as ICR[4R+] mice. As shown in Supplementary Fig. 1b, both ICR[4R+] and ICR background parental mice were infected with HCV. We detected that serum HCV particle copies, liver HCV RNA positive (+) and negative (−) strand replication continuously increased and peaked at Day 42, and then maintained at high levels at least during our detection period (for 56 days) in ICR[4R+] mice but not in HCV-infected parental ICR mice (Supplementary Fig. 1c−e). The liver function test alanine transaminase (ALT) levels (indicating the level of liver damage) also significantly increased at Day 49 post infection in ICR[4R+] mice but not in HCV-infected parental ICR mice (Supplementary Fig. 1f). Immunohistochemistry results also showed that HCV Core protein expression was observed at Day 49 post infection in livers from ICR[4R+] mice but not in HCV-infected parental ICR mice (Supplementary Fig. 1g). ICR[4R+] transgenic mice were either infected with HCV or mock infected with PBS. In agreement with the above result (Fig. 1a), the mRNA level of *Fut8* was also remarkably increased in hepatocytes of ICR[4R+] mice infected with HCV than those of uninfected ICR[4R+] mice at Day 40 post infection (Fig. 1c).

To identify the viral components responsible for upregulating FUT8 expression, we constructed a panel of eukaryotic expression plasmids encoding all 10 HCV proteins (core, P7, E1, E2, NS2, NS3, NS4a, NS4b, NS5a, and NS5b) with C-terminal His-epitope tags, and transfected them into Huh7 cells to screen for their ability to upregulate FUT8 expression. Among the 10 HCV gene expression plasmids, pcDNA3.1-E2 transfection caused the most significant upregulation of FUT8 protein expression (Fig. 1d), and the dual-fluorescence reporter gene experiment also showed that *FUT8* promoter could only be activated by enveloped protein E2 (Fig. 1e) in Huh7 cells. E2 upregulated FUT8 expression and core fucosylation in an E2 dose-dependent manner as shown in Fig. 1f, g. E2 is the main envelope protein of HCV that mediates binding to receptors (such as CD81) on human hepatocytes during viral infection[37]. These results suggest that HCV envelope protein E2 is critical for HCV-induced FUT8 expression.

Interestingly, we also observed increased FUT8 expression induced by other enveloped viral infections, including VSV (Fig. 1h and Supplementary Fig. 2a), HSV-1 (Supplementary Fig. 2b) and SARS-CoV-2 (Supplementary Fig. 2c) infection, but not by SeV (Supplementary Fig. 2d) or *E. coli* infection (Supplementary Fig. 2e). We also investigated whether other enveloped viral membrane proteins increased FUT8 expression. We found that viral envelope proteins, including VSV-G (Fig. 1i), SARS-CoV-2-Spike (S) (Supplementary Fig. 2f), and HIV-gp120 (Supplementary Fig. 2g), increased FUT8 mRNA and protein expression in HEK293T cells. Furthermore, increased core fucose was detected by AAL blotting in 293T cells transfected with VSV-G, SARS-CoV-2-S, and HIV-gp120 expression plasmids (Fig. 1j and Supplementary Fig. 2f, g).

In general, cellular fucosylation is regulated by GDP-fucose and GDP-fucose transporter, in addition to FUTs[38]. We further examined the expression of GDP-mannose 4, 6-dehydratase (GMDS, catalyzing the first step in the synthesis of GDP-fucose from GDP-mannose), GDP-4-keto-6-deoxy-mannose-3, 5-epimerase-4-reductase (FX), solute Carrier Family 35 Member C1 (SLC35C1, a GDP-fucose transporter) after HCV/VSV infection by RT-qPCR. We found that no significant differences were observed for the expression of GMDS, FX, SLC35C1 in HCV-infected Huh7 cells or VSV-infected HEK293T cells (Supplementary Fig. 2h−m). These findings suggest that virus-induced

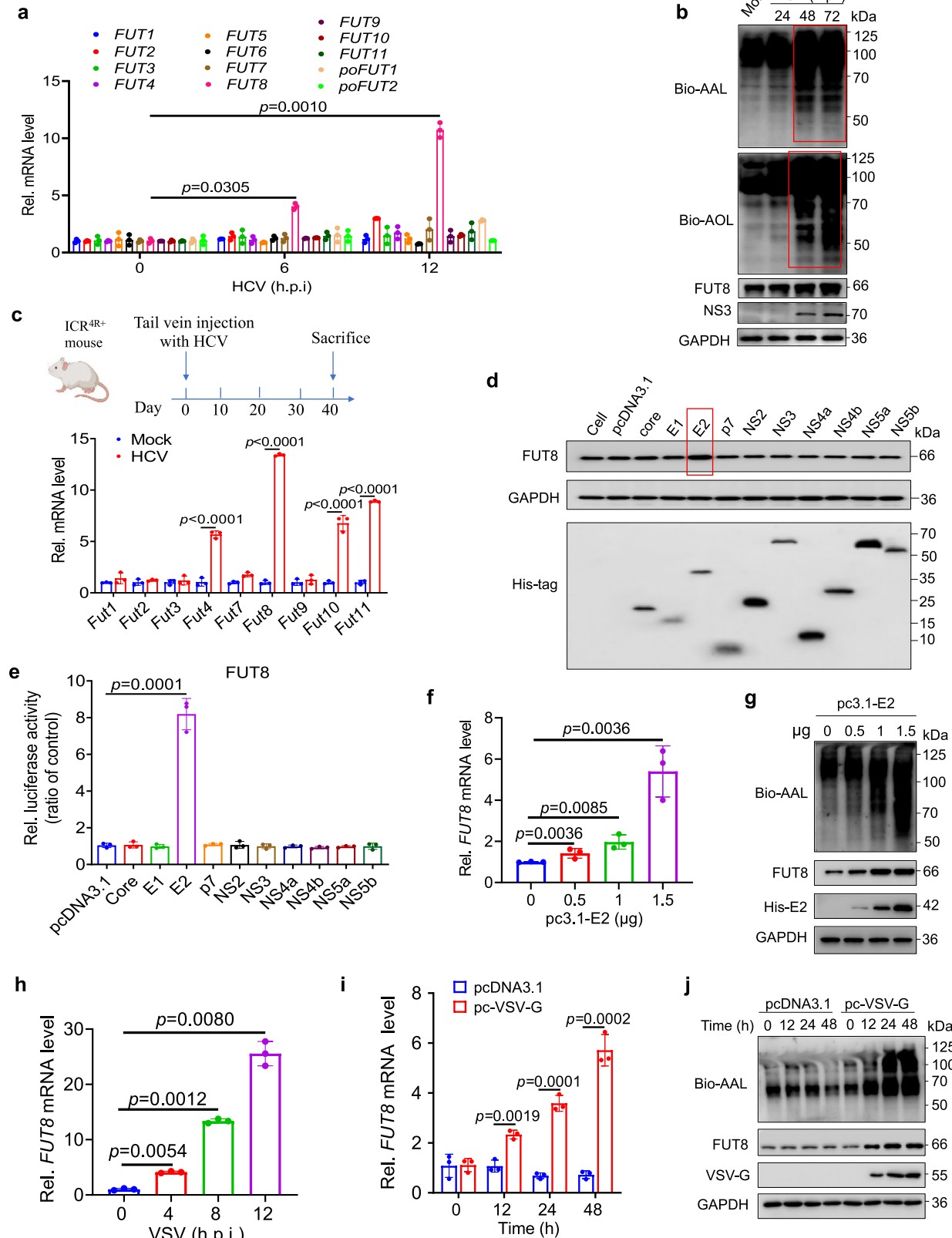

core fucosylation upregulation is mainly induced by FUT8, but not regulated by GDP-fucose and GDP-fucose transporter.

Collectively, these results provide evidence that FUT8 is upregulated by multiple enveloped viruses and viral envelope proteins, and the results from the mouse models confirm that FUT8 expression is also upregulated in primary cells after HCV infection.

**HCV infection and HCV-E2 manipulate host transcription factor SNAIL to induce FUT8 expression**

Next, we explored the detailed mechanism underlying the increase in FUT8 expression caused by HCV infection. We examined the host transcription factors that were manipulated by the viruses to induce FUT8 expression.

**Fig. 1 | HCV and VSV infection upregulates cellular *FUT8* mRNA and protein expression. a** Analysis of FUT family mRNA expression in Huh7 cells infected with HCV (MOI = 0.1) for the indicated time using RT-qPCR (*n* = 3 per group per study). **b** Lectins (AAL and AOL) and immunoblot analysis of core fucosylation and FUT8 in lysates of Huh7 cells infected with HCV (MOI = 0.1) for the indicated time. **c** RT-qPCR analysis of Fut family mRNA expression in hepatocytes of ICR[4R+] mice infected with HCV (1 ×10[6] copies per mouse) in vivo for 40 d (*n* = 3 mice per group per study). The figure is created with Biorender.com. **d** Immunoblot analysis of FUT8 in lysates of Huh7 cells transiently transfected with eukaryotic expression plasmids encoding each of the ten HCV proteins. **e** *FUT8* promoter activity in Huh7 cells transiently co-transfected with pGL3 luciferase reporter system and eukaryotic expression plasmids of individual HCV genes. Luciferase activities were analyzed as fold induction (*vs.* pcDNA3.1 empty vector group) (*n* = 3 per group per study). **f** RT-qPCR analysis of *FUT8* mRNA expression in Huh7 cells transiently transfected with His-vector or His-E2 plasmids at the indicated concentrations (*n* = 3 per group per study). **g** Immunoblot analysis of FUT8 and core fucosylation in lysates of Huh7 cells transiently transfected with indicated plasmids at the indicated concentrations. **h** RT-qPCR analysis of *FUT8* mRNA expression in HEK293T infected with VSV (MOI = 0.1) for the indicated time (*n* = 3 per group per study). RT-qPCR analysis of *FUT8* mRNA expression (**i**) and lectin and immunoblot analysis of FUT8 and core fucosylation (**j**) in lysates of HEK293T cells transiently transfected with the indicated plasmids for the indicated time. Data in all quantitative panels are presented as mean ± SD. Data are representative of three independent experiments. Data are normalized based on human *GAPDH* for **a**, **e**, **f**, **h**, **i** and mouse *Gapdh* for **c**. Two-tailed unpaired Student's *t* test was used to assess the statistical difference in **a** and **h** (*vs.* 0 h.p.i.), **c** (*vs.* Mock), **e** and **i** (*vs.* pcDNA3.1), **f** (*vs.* 0 μg). Source data are provided as a Source Data file.

The SNAIL protein acts as a critical transcription factor for FUT8 expression in breast cancer[39]; however, it is unclear whether SNAIL serves as a transcription factor for FUT8 during viral infection. We used the electrophoretic mobility shift assay (EMSA) to detect the direct binding of SNAIL to the *FUT8* promoter during HCV infection[39]. We synthesized a 36 nucleotide DNA probe corresponding to *FUT8* promoter region containing the SNAIL transcription factor-binding region (Fig. 2a). As the concentration of SNAIL protein increased, it bound more tightly to the WT-Fam probe of the *FUT8* promoter (SNAIL binding region (−295 to −260): GGCAGGTGAGA, indicated by red font in Fig. 2a, lanes 2–5). When the SNAIL protein was pretreated with a 10-fold excess of unlabeled "competitor" probes (300 nM), the binding of SNAIL protein to WT-Fam probe was abolished (Fig. 2a, lanes 6 *vs.* 5). However, when mutant probes were used, SNAIL lost its ability to bind to the mutant probes (Fig. 2a, lane 7 *vs.* 5). Furthermore, we found that SNAIL overexpression upregulated the expression of *FUT8* mRNA (Fig. 2b) and induced *FUT8* promoter activation in a dual-fluorescence reporter gene experiment (Supplementary Fig. 3a). Furthermore, cytoplasmic SNAIL decreased and nuclear SNAIL increased at 24 h post-transfection with pcDNA3.1-Myc-His-E2 in Huh7 cells by nucleus-cytoplasm-fractionation assay (Fig. 2c), suggesting that HCV-E2 induced the nuclear translocation of the transcription factor SNAIL in Huh7 cells. These data strongly indicate that HCV and HCV-E2 induce FUT8 expression through SNAIL nuclear translocation and recruit SNAIL to the *FUT8* promoter in Huh7 cells.

## HCV infection and HCV-E2 upregulate FUT8 via EGFR-AKT-SNAIL activation

Many viral infections (e.g., HCV, VSV) stimulate the EGFR/AKT endocytosis signaling pathways[40–43]. EGFR/AKT activation results in SNAIL protein expression[44]. Next, we explored the signaling pathway involved in HCV-EGFR/AKT-SNAIL-mediated upregulation of FUT8. We found that p-EGFR was activated at 1 h post HCV infection, reaching its peak at 2 h (Fig. 2d). p-AKT was initially activated at 2 h, and began to decrease at 6 h. The transcription factor SNAIL began to increase at 6 h. Subsequently, an increase in FUT8 expression was observed at 48 h (Fig. 2d). This data indicate that HCV infection causes HCV E2-EGFR-p-AKT-SNAIL-FUT8 axis activation (Fig. 2d). Viral infection-induced FUT8 expression was dependent on EGFR, AKT, and SNAIL, as HCV-infection-induced upregulation of *FUT8* mRNA levels (Fig. 2e) and *FUT8* promoter activation (Fig. 2f) were attenuated in Huh7 cells after silencing EGFR, AKT, or SNAIL. Using western blotting (WB) and AAL lectin blot assays, we also observed that HCV infection could induce an increase in FUT8 expression and core fucosylation, but this promoting effect disappeared after knockdown of EGFR (Fig. 2g), AKT (Fig. 2h), or SNAIL (Fig. 2i) in HCV-infected Huh7 cells at the indicated time points. These data strongly suggest that HCV infection upregulates FUT8 transcription through EGFR-AKT-SNAIL activation.

The binding of HCV particles and HCV-E2 proteins to human hepatocytes induces EGFR activation[45]. However, whether exogenous HCV-E2 in cells interaction with EGFR remains unknown. Full-length HCV-E2 contains transmembrane region[46,47], usually could be expressed on the cell surface. Confocal microscopy analysis showed that HCV-E2 (red) co-localized with cellular endogenous EGFR (green) on the cell surface of Huh7 cells transfected with pcDNA3.1-myc-His-E2 (co-localization indicated by the orange color in Supplementary Fig. 3b), but not in pcDNA3.1 empty vector group. To further assess whether E2 and EGFR likely interact when viral particles are in contact with the cell surface during the entry process of HCV infection, we performed HCV infection of Huh7 cells for different time courses (5 min, 15 min, 1 h and 48 h). We found that HCV-E2 interacted with EGFR after 15 min upon infection, and both HCV-E2 and EGFR transferred from the cell surface into the cell interior (inducing EGFR internalization) by both confocal microscopy (Supplementary Fig. 3c) and flow cytometry analysis (Supplementary Fig. 3d, e). And we also found that, at 48 h post infection, the majority of EGFR and a small fraction of E2 ended up on the cell surface, while the majority of E2 was intracellular (Supplementary Fig. 3c−e). These results suggest that the HCV-E2-EGFR colocalization might induce EGFR activation.

We further compared the effects of different viral envelope proteins (HCV-E2, VSV-G, SARS-CoV-2-spike, and HIV-gp120) on EGFR-AKT-SNAIL-FUT8 pathway activation. Both WT and EGFR KO Huh7 cells were transfected with indicated expression plasmids (Supplementary Fig. 3f, g), and the results showed that these viral envelope proteins all induced FUT8, core fucosylation (assessed by AAL lectin blot) and SNAIL upregulation in WT but not in EGFR KO cells (Supplementary Fig. 3f, g). Among these viral envelope proteins, VSV-G induced the highest level of FUT8, core fucosylation and SNAIL expression at 48 h post transfection (Supplementary Fig. 3f), followed by SARS-CoV-2-Spike, HCV-E2 and HIV-gp120. We also found that these viral envelope proteins induced upregulation of p-EGFR and p-AKT in WT at 12 h post transfection, but not in EGFR KO cells (Supplementary Fig. 3g). These results suggest that all these viral envelope proteins could induce EGFR-AKT-SNAIL-FUT8 pathway, but with different degrees of activation.

In addition, Sendai virus (SeV) is also an enveloped virus, but we observed that SeV could not induce FUT8 mRNA expression (Supplementary Fig. 2d). We assessed whether SeV activates EGFR and found that SeV infection had no effect on core fucosylation, FUT8 expression, or p-EGFR-SNAIL activation compared to the mock-infected group (Supplementary Fig. 3h). This result suggests that SeV, albeit an enveloped virus, does not induce FUT8 expression, which may be attributable to the lack of EGFR activation by SeV infection.

## HCV-upregulated FUT8 further activates the core fucosylated-EGFR and its downstream cascade EGFR-JAK1-STAT3

Next, we explored that HCV induced-FUT8 affected and modified critical molecules in host cells. Previous reports have suggested that EGFR is involved in HCV infection and that FUT8 can promote EGFR

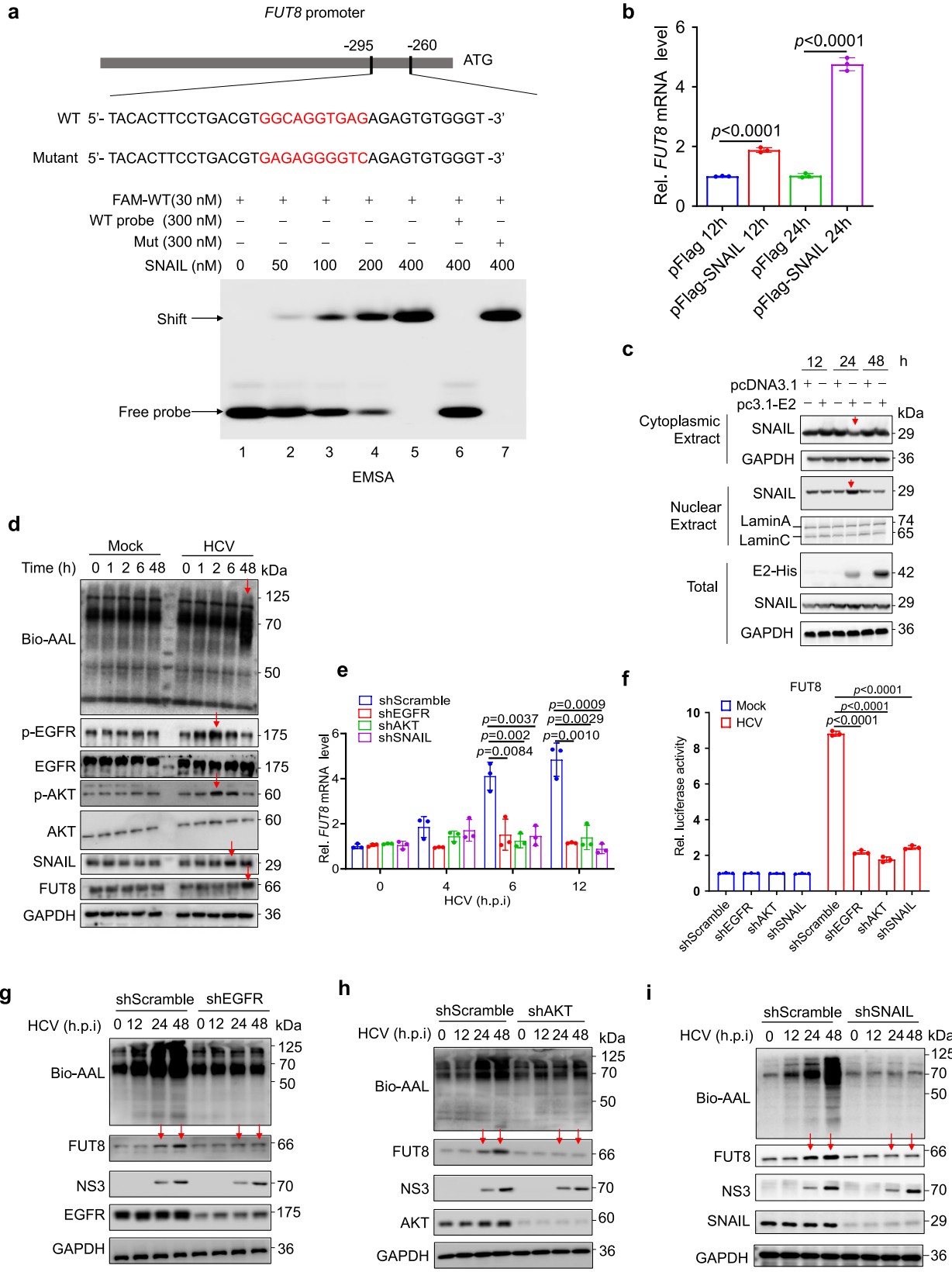

dimerization and phosphorylation in lung cancer cells[22,48]. However, whether viral infections induce EGFR core fucosylation remains unclear. Therefore, we evaluated whether virus-upregulated FUT8 could activate core fucosylated EGFR in the target cells. As shown in immunofluorescence microscopy analysis (Fig. 3a), the expression of FUT8 (Blue) was increased in the HCV-infected cells compared to that

of the PBS-treated cells. HCV-infected cells displayed a dispersed Golgi pattern and EGFR internalization. Additionally, HCV infection increased EGFR (Green) colocalization with FUT8 (Blue) and Golgi marker GM130 (Red). The overlapping regions of the red, green and blue generated white images (indicated by purple arrows in the images).

**Fig. 2 | HCV infection and its envelope protein E2 upregulate FUT8 through EGFR-AKT-SNAIL activation and SNAIL nuclear translocation. a** EMSA of SNAIL binding to the promoter of FUT8. The purified recombinant His-SNAIL protein was incubated with probes, and the protein-DNA complexes were separated on native polyacrylamide gels. "−" represents absence and "+" represents presence. Unlabeled or mutant probes at 300 nm were added to the reaction mixture for competition and testing binding specificity. Arrows indicate the positions of shifted bands. **b** RT-qPCR analysis of *FUT8* mRNA expression in Huh7 cells transiently transfected with Flag-vector or Flag-SNAIL plasmids for the indicated time (*n* = 3 per group per study). **c** Immunoblot analysis of the subcellular fraction of Huh7 cells expressing HCV-E2 for 0–48 h. **d** Lectin and immunoblot analysis of EGFR, AKT, SNAIL, and core fucosylation in lysates of Huh7 cells infected with HCV (MOI = 0.1) for the indicated time. **e** RT-qPCR analysis of *FUT8* mRNA expression in

Huh7 cells transfected with EGFR shRNA, AKT shRNA, or SNAIL shRNA for 48 h, and then infected with HCV for the indicated time (*n* = 3 per group per study). **f** *FUT8* promoter activity in Huh-7 cells transfected with EGFR shRNA, AKT shRNA, or SNAIL shRNA for 48 h, and then infected with HCV for 12 h, detected using the luciferase reporter assay. Luciferase activity was analyzed as fold induction (*n* = 3 per group per study). Immunoblot analysis of FUT8, core fucosylation, and NS3 in lysates of Huh7 cells transfected with EGFR shRNA (**g**), AKT shRNA (**h**), or SNAIL shRNA (**i**) for 48 h, and then infected with HCV (MOI = 0.1) for the indicated time. Data in all quantitative panels are presented as mean ± SD. Data are representative of three independent experiments. Data are normalized based on *GAPDH* for **b** and **e**. Two-tailed unpaired Student's *t* test was used to assess the statistical difference in **b** (*vs*. pFlag), **e** and **f** (*vs*. shScramble). Source data are provided as a Source Data file.

We also found that EGFR core fucosylation was increased by HCV infection and FUT8 overexpression through the AAL-lectin assay (Fig. 3b, lane 2 *vs*. 1) and decreased after treatment with the FUT8 inhibitor 2FF (a competitive inhibitor of FUT8 as a GDP derivative[49]) (Fig. 3b, lane 5 *vs*. 2) treatment. Similar to the pattern of EGFR core fucosylation, the levels of EGFR phosphorylation were also increased in FUT8-overexpressed Huh7 cells but decreased in FUT8 inhibitor 2FF-treated Huh7 cells after HCV infection (Fig. 3b, lane 5 *vs*. 2), suggesting that EGFR core fucosylation enhanced EGFR phosphorylation during HCV infection. Furthermore, FUT8 overexpression or epidermal growth factor (EGF, EGFR-ligand) administration led to an increase in JAK1 and STAT3 phosphorylation, decrease in RIG-I and p-IRF3 in Huh7 cells, but treatments with FUT8 inhibitor 2FF or EGFR phosphorylation inhibitor erlotinib displayed opposite effects (Fig. 3c). These results suggest that FUT8 induces rapid phosphorylation and activation of EGFR, JAK1, STAT3, and downregulation of RIG-I/p-IRF3.

Next, we investigated the N-glycosylation sites of EGFR, which are critical for the activation of downstream signaling pathways. EGFR contains 14 N-glycosylation sites, as predicted by the Asn-X-Ser/Thr sequence; therefore, we constructed 14 N-glycosylation site-specific mutant plasmids of EGFR (Fig. 3d and Supplementary Fig. 4a). We constructed EGFR KO Huh7 cells and complemented them with WT or each of the 14 N-glycosylation site-specific mutant EGFR. We found that only N352Q site-specific mutation dampened JAK1 and STAT3 phosphorylation in Huh7 cells (Fig. 3d).

Above results suggest that HCV infection induced-FUT8 led to EGFR core fucosylation upregulation. N-glycosylation site of EGFR at N352AT plays a key role in the fucosylated-EGFR-JAK1-STAT3 signaling pathway.

## FUT8-induced pSTAT3 activation recruits Trim40 and promotes RIG-I K48-linked ubiquitination and proteasomal degradation
The JAK1-STAT3 signaling pathway is a major pathway which is activated by EGFR family members[50,51]. The STAT3-mediated signaling pathway regulates the immune response to IFNs[52]. We investigated whether the JAK1-STAT3 pathway modulates RIG-I expression. We found that FUT8 overexpression reduced RIG-I expression and IRF3 phosphorylation after Huh7 cells infected with HCV, but treatment with 2FF or erlotinib had the opposite effect (Fig. 3c).

As shown in Supplementary Fig. 4b, RIG-I protein levels were much lower from 12 h to 24 h after transfection with STAT3 in Huh7 cells than in control cells. In contrast, STAT3 overexpression did not affect *RIG-I* mRNA levels (Supplementary Fig. 4c), suggesting that STAT3 may mediate RIG-I ubiquitination and degradation. It has been reported that TRIM40 promotes proteasomal degradation by conjugating K48-linked ubiquitin chains on RIG-I[53]. To confirm whether STAT3 is involved in the stabilization of RIG-I by Trim40, we transfected Huh7.5.1, cells with RIG-I, HA-ubiquitin, STAT3 expression plasmids, or Trim40 siRNA (siTrim40). Ubiquitin overexpression induced mild RIG-I degradation (Fig. 4a, lane 3 *vs*. 2). In STAT3

overexpressing cells, K48-linked ubiquitination and degradation of RIG-I further increased (Fig. 4a, lane 5 *vs*. 4); however, siTrim40 protected RIG-I from ubiquitination (Fig. 4a, lanes 7 *vs*. 6). We also found that K48-linked ubiquitination and degradation of RIG-I increased in FUT8 overexpressing cells; however, siTrim40 protected RIG-I from ubiquitination and degradation (Fig. 4b).

We hypothesized that FUT8 may affect the interaction between Trim40 and RIG-I. To test this hypothesis, RIG-I and Trim40 were immunoprecipitated from Huh7.5.1 cells. RIG-I and Trim40 interactions were detected in FUT8 overexpression cells; however, RIG-I and Trim40 interactions decreased in 2FF treated cells (Fig. 4c). Collectively, these results revealed that FUT8 degrades RIG-I protein expression by promoting the interaction between RIG-I and Trim40.

## FUT8 promotes HCV RNA replication through the core fucosylated EGFR and p-EGFR-p-JAK1-p-STAT3 pathway
Next, we assessed the effect of the interaction between FUT8 and the EGFR-JAK1-STAT3 signaling pathway on HCV infection. FUT8 overexpression promoted HCV RNA replication dependent on the EGFR-JAK1-STAT3 signaling pathway (Fig. 4d, e), as indicated by the significant decrease in HCV RNA replication (Fig. 4d) and HCV NS3 protein expression (Fig. 4e) after knockdown of EGFR, JAK1, and STAT3, respectively. In contrast, exogenous FUT8 overexpression still promoted HCV RNA replication in cells co-transfected with AKT small hairpin RNA (shRNA) (Supplementary Fig. 4d), suggesting that FUT8 promotes HCV RNA replication through EGFR-JAK1-STAT3 signal, but not through the EGFR-AKT pathway.

Time course experiment showed that HCV infection led to RIG-I increase at early stage (8 h.p.i.) and FUT8/core fucosylation upregulation at late stage (12–72 h.p.i.). Subsequently, a decrease in RIG-I protein level was observed after 48–72 h.p.i., since increased-FUT8 induced RIG-I degradation at late stage (Fig. 4f). But the RIG-I protein level at 72 h.p.i. was still higher than that at 0 to 4 h.p.i., due to HCV infection. This result suggests that endogenous RIG-I could be induced by HCV infection at early stage (8 h.p.i.), and then partly degraded due to the HCV-induced FUT8 expression at late stage (48–72 h.p.i.).

These above results revealed the relationship between HCV E2-EGFR-p-AKT-SNAIL-FUT8 pathway and fucosylated-EGFR-p-JAK-p-STAT3 pathway. As mentioned above, HCV engaged EGFR to activate p-AKT-SNAIL pathway at early stage (Fig. 2d, p-AKT peaked at 2 h.p.i. and remarkably decreased at 48 h.p.i.), thus induced FUT8 expression (Fig. 2d and Supplementary Fig. 4e, at 24–72 h.p.i.). Subsequently upregulated-FUT8 induced fucosylated-EGFR-p-JAK-p-STAT3 activation after 24 h.p.i. (Fig. 3c and Supplementary Fig. 4e). Interestingly, this promoting effect disappeared after knockdown of AKT (Supplementary Fig. 4e). We also demonstrated that FUT8 inhibitor 2FF could suppress the fucosylated-EGFR-p-JAK-p-STAT3 activation (Fig. 3c), suggesting that increased-FUT8 is the initiator of fucosylated-EGFR-p-JAK-p-STAT3 pathway. So, our results strongly suggest that HCV E2-p-EGFR-p-AKT-SNAIL-FUT8 pathway is at the upstream of fucosylated-EGFR-p-JAK-p-STAT3 pathway.

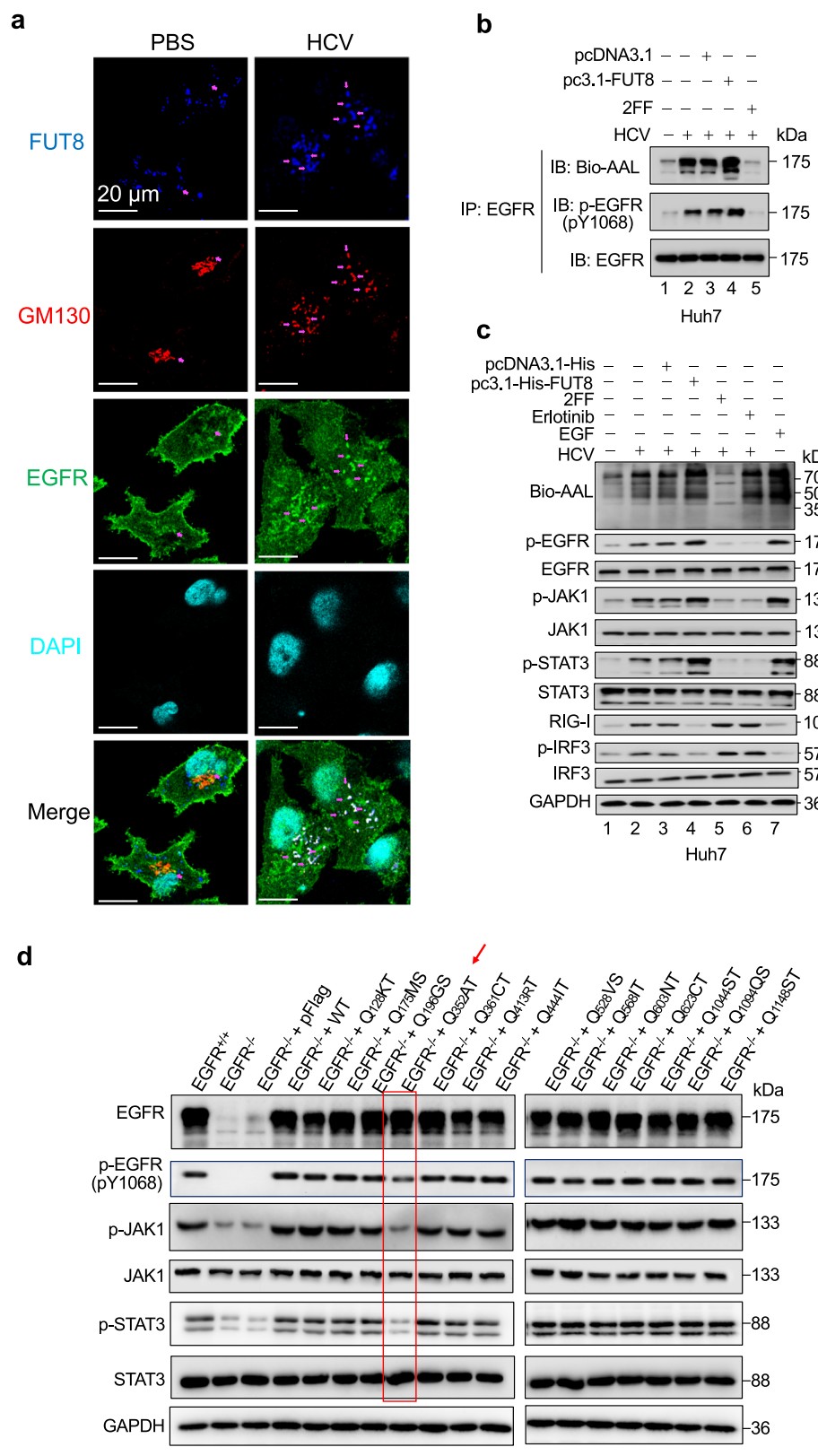

Consistent with these findings (Fig. 4d, e), treatment with erlotinib blocked FUT8 overexpression-induced HCV RNA replication (Fig. 4g, lane 6 *vs.* 4). Moreover, EGFR KO significantly decreased HCV RNA replication compared to that in WT Huh7 cells (Fig. 4h). However, complementation with WT EGFR or the 13 N-glycosylation site-specific mutants rescued HCV RNA replication, except for EGFR N352Q mutant

(Fig. 4h), suggesting that FUT8 overexpression impairs the host anti-viral response through the core fucosylated EGFR, and EGFR N-glycosylation mutation (Q352AT) suppresses HCV RNA replication.

To further confirm whether N-glycosylation mutations affect EGFR localization, Huh7 cells were transfected with EGFR WT or N-glycosylation mutations plasmids. The cells were treated with 10 nM

**Fig. 3 | EGFR N-glycosylation and phosphorylation are required for FUT8-induced effects. a** Huh7 cells were infected with HCV (MOI = 0.1) for 48 h. Cells were fixed and labeled for EGFR (green), FUT8 (blue) and the Golgi marker GM130 (red). DAPI (cyan color) was used to stain nuclei. Purple arrows indicate the colocalization of EGFR with FUT8 and Golgi. Representative confocal microscopy images are shown. **b** Core fucosylation of EGFR affects tyrosine phosphorylation (p-Tyr) of EGFR. His-FUT8-overexpressing (transfection for 48 h) or 2FF-treated (for 48 h) Huh7 cells were infected with HCV (MOI = 0.1) for 24 h. Immunoblot analysis of p-EGFR (pY1068) in cell lysates immunoprecipitated with antibody to EGFR.

**c** Immunoblot analysis of phosphorylated (p-) or total proteins in lysates of Huh7 cells transiently transfected with His-vector or His-FUT8 plasmids or treated with 2FF, Erlotinib or EGF for 48 h, and then infected with HCV for 24 h. **d** Immunoblot analysis of phosphorylated (p-) or total EGFR/JAK1/STAT3 in lysates of Huh7 cells transiently transfected with Flag-vector, Flag-EGFR, or EGFR N-glycosylation deletion mutant plasmids for 48 h followed by stimulation with 10 nM EGF for 1 h. Data are representative of three independent experiments. Source data are provided as a Source Data file.

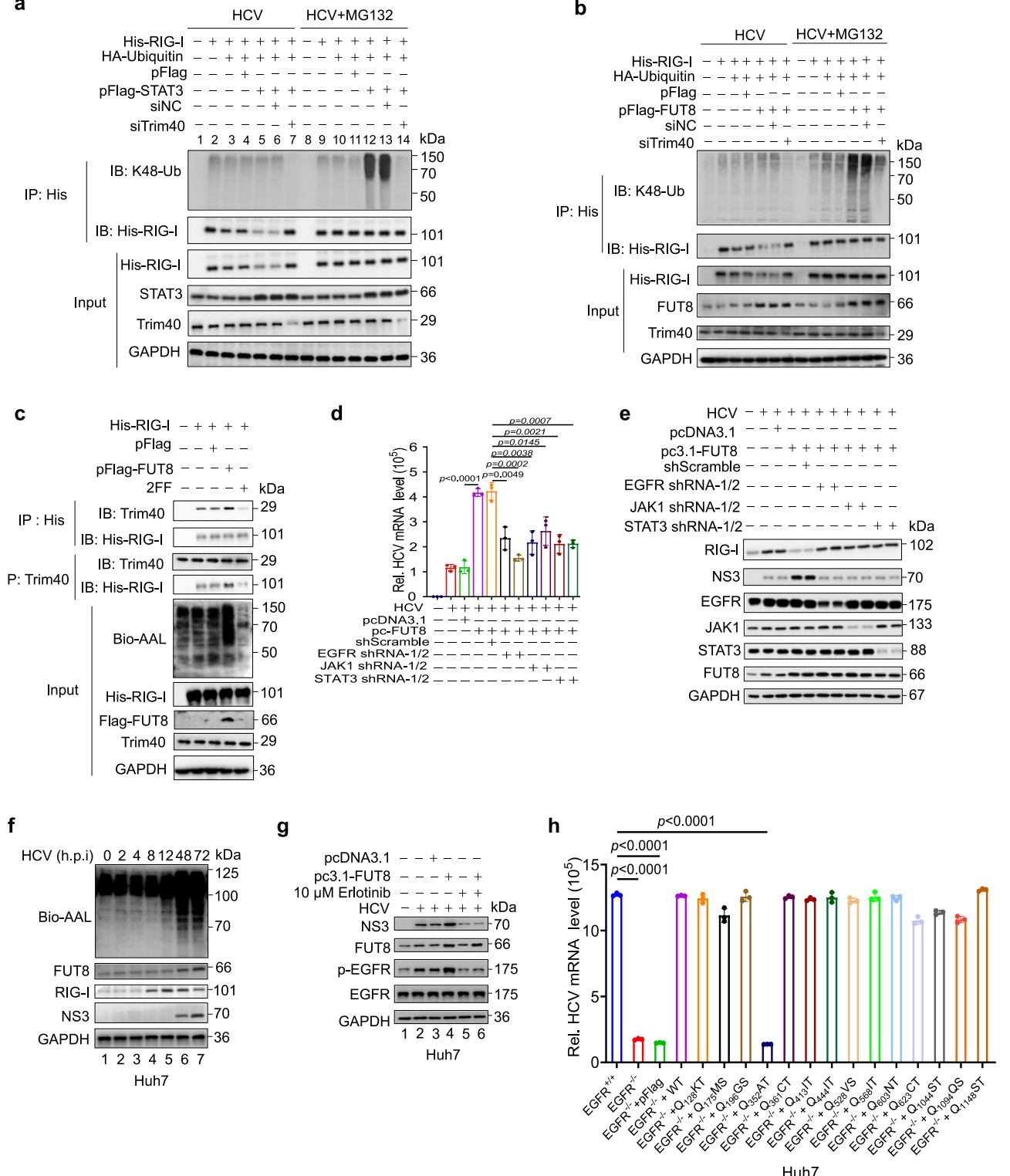

**Fig. 4 | FUT8-induced pSTAT3 activation recruits Trim40 and promotes RIG-I K48-linked ubiquitination and proteasomal degradation. a, b** Huh7.5.1 cells were transfected with indicated plasmids or siTrim40 (50 nM) for 24 h, and then infected with HCV (MOI = 0.1) for 48 h in the presence or absence of MG132 (10 mM). Cells were harvested for immunoblot analysis of K48-Ub immunoprecipitated with antibody to His tag. **c** Huh7.5.1 cells were transfected with indicated plasmids or treated with 2FF (100 µM) for 24 h, and then infected with HCV (MOI = 0.1) for 48 h in the presence of MG132 (10 mM). Cells were harvested for immunoblot analysis of Trim40 with antibody to His-RIG-I or His-tag with antibody to Trim40. RT-qPCR analysis of HCV mRNA expression (**d**) and immunoblot analysis of FUT8, RIG-I, NS3, EGFR, JAK1, and STAT3 (**e**) in Huh7 cells transiently transfected with His-vector or His-FUT8 plasmids or co-transfected with EGFR shRNA, JAK1 shRNA, or STAT3 shRNA for 48 h, and then infected with HCV (MOI = 0.1) for 48 h (**d**, n = 3 per group per study) or 72 h (**e**). **f** Lectin blot analysis

for core fucosylation and immunoblot analysis for FUT8/RIG-I/NS3 in HCV-infected Huh7 cells (MOI = 0.1) for the indicated time. **g** Immunoblot analysis of phosphorylated (p-), total EGFR, FUT8 and NS3 in lysates of Huh7 cells transiently transfected with His-vector or His-FUT8 plasmids or treated with 10 µM Erlotinib for 48 h, and then infected with HCV (MOI = 0.1) for 72 h. **h** RT-qPCR analysis of HCV mRNA expression in Huh7 cells with transiently transfected with Flag-vector, Flag-EGFR, or EGFR N-glycosylation deletion mutant plasmids for 48 h, and then infected with HCV (MOI = 0.1) for 48 h. Data are normalized based on *GAPDH* for **d** and **h** (n = 3 per group per study). Data in all quantitative panels are presented as mean ± SD. Data are representative of three independent experiments. One-way ANOVA followed by Sidak's or Dunnett's multiple comparisons test was used to assess the statistical difference for d and h. Source data are provided as a Source Data file.

EGF and then labeled for EGFR and the lysosomal marker Lamp1. EGFR N352Q mutation inhibited the ability of EGF to stimulate EGFR endocytosis and colocalization with Lamp1 (Supplementary Fig. 4f). These data suggest that EGFR N-glycosylation site (N352AT) may be required for EGFR endocytosis and internalization.

Collectively, above results suggest that FUT8 promotes HCV RNA replication through the core fucosylated EGFR and p-EGFR-p-JAK1-p-STAT3 signaling pathways. EGFR N-glycosylation site (N352AT) is required for EGFR endocytosis and HCV RNA replication.

## FUT8 promotes HCV RNA replication through suppression of cellular RIG-I-induced type I IFN production

Since we found that FUT8-induced pSTAT3 activation recruits Trim40 and promotes RIG-I K48-linked ubiquitination and proteasomal degradation (Fig. 4a), we further characterized the roles of FUT8 in viral RNA replication and RIG-I-induced IFN-I production. We found that overexpression of FUT8 significantly increased HCV RNA replication (Fig. 5a, lane 8 *vs.* 7), whereas FUT8 knockdown exhibited the opposite effects in Huh7 cells (Fig. 5a, lane 10 *vs.* 9). However, overexpression or knockout of FUT8 (FUT8 KO) in Huh7.5.1 cells showed no effect on HCV RNA replication due to the deficiency of RIG-I in Huh7.5.1 cells (Fig. 5b, lane 3-6 *vs.* 1-2). Since Huh7.5.1 cells, but not Huh7 cells, carry the T55I mutation in the first CARD of RIG-I, which disrupts the RIG-I-MAVS axis signaling pathway[29,54], Huh7.5.1 cell line was thus more adaptable to imitate tree shrew cells, which have a natural RIG-I deficiency[55]. Given this difference, we overexpressed exogenous RIG-I in Huh7.5.1 cells, HCV RNA replication was significantly suppressed (Fig. 5b, lane 7 *vs.* 1). Overexpression of FUT8 in Huh7.5.1 cells complemented with exogenous RIG-I increased HCV RNA replication (Fig. 5b, lane 9 *vs.* 8), whereas FUT8 KO Huh7.5.1 cells complemented with exogenous RIG-I decreased HCV RNA replication (Fig. 5b, lane 10 vs. 7). In FUT8 KO Huh7.5.1 cells complemented with exogenous FUT8 and RIG-I, HCV RNA replication increased in contrast to that in FUT8KO Huh7.5.1 cells complemented with exogenous RIG-I alone (Fig. 5b, lane 12 *vs.*11). Similar results were observed for HCV NS3 protein expression (Fig. 5c), as indicated by increased NS3 protein expression after overexpression of FUT8 and exogenous RIG-I in Huh7.5.1 cells (Fig. 5c, lane 9 *vs.* 8) and decreased NS3 protein expression in FUT8 KO Huh7.5.1 cells complemented with exogenous RIG-I (Fig. 5c, lane 10 *vs.* 7). In Huh7.5.1 cells, FUT8 overexpression or knockdown had no effects on IFN-β mRNA expression (Fig. 5d). However, FUT8 overexpression could inhibit IFN-β expression and promoted HCV RNA replication in Huh7.5.1 cells complemented with exogenous RIG-I (Fig. 5b, d).

RIG-I triggers the anti-viral IFN-I immune response upon the detection of viral RNA[56]; therefore, we assessed the effect of FUT8 on virus-induced IFN-I production. We found that FUT8 overexpression inhibited *IFN-β* mRNA expression (Fig. 5e, lane 8 *vs.* 7) and protein

expression (Fig. 5f, lane 8 *vs.* 7) in Huh7 cells carrying endogenous RIG-I. Treatment with the FUT8 inhibitor 2FF induced higher *IFN-β* mRNA (Fig. 5e, lane 6 *vs.* 5) and protein expression (Fig. 5f, lane 6 *vs.* 5) in HCV-infected Huh7 cells. The effects of FUT8 in RIG-I-rescued Huh7.5.1 cells (Fig. 5b, c, d) were consistent with those in Huh7 cells (Fig. 5a, e, f). These data suggest that FUT8 promotes HCV RNA replication through suppression of cellular RIG-I-induced IFN-I production.

## FUT8 promotes other RNA viral replication, but not DNA viral replication, through suppression of IFN-I production

Next, we determined the effects of FUT8 on IFN-I induced by other RNA viruses (VSV and SeV). We observed similar results for the effects of FUT8 on the replication of other RNA viruses such as VSV (Fig. 5g and Supplementary Fig. 5a) and SeV (Fig. 5h); FUT8 overexpression promoted viral RNA replication, whereas FUT8 KO suppressed viral RNA replication. We also found that VSV and SeV RNA levels were significantly lower in 2FF-treated HEK293 cells than in the control cells (Fig. 5g, h).

HCV is a single-positive-stranded RNA virus that produces negative-stranded RNA as replication intermediates[57]. Thus, the detection of negative-stranded RNA in infected cells is considered a marker of active viral replication. We tested for the presence of negative-stranded HCV RNA using Tth-based strand-specific RT-qPCR in indicated cells. We observed FUT8 effects on HCV negative-strand replication in Huh7 cells. Overexpression of FUT8 promoted the production of negative-stranded HCV RNA (Fig. 5i, lane 5 *vs.* 4), whereas the FUT8 inhibitor 2FF had the opposite effect (Fig. 5i, lane 3 *vs.* 2). These above findings demonstrate that FUT8 may facilitate RNA viral replication.

FUT8 KO led to a significant increase *IFN-β* mRNA expression after VSV or SeV infection (Fig. 5j), but not after DNA virus HSV-1 infection (Supplementary Fig. 5b). In contrast, in FUT8 overexpression (Fig. 5k) or FUT8 rescued (Fig. 5l) led to significantly decreased *IFN-β* mRNA expression after VSV or SeV infection (Fig. 5k, l), but not after HSV-1 infection (Supplementary Fig. 5b).

Albeit SeV could not activate the upstream p-EGFR-SNAIL-FUT8 pathway (Supplementary Fig. 3h), FUT8 overexpression could activate subsequent core fucosylated EGFR-p-JAK1-p-STAT3 pathways, and thus lead to RIG-I degradation, IFN-I downregulation and RNA viral replication (including SeV) (Fig. 5h, j–l).

As RNA viruses activate the RIG-I-IRF3-IFN-I pathway, while DNA viruses engage the cGAS-STING-IRF3-IFN-I pathway. DNA virus HSV-1 infection induced upregulation of IFN-β dependent on cGAS-STING-IRF3 pathway[58], we further assessed the effects of FUT8 on the HSV-1-induced cGAS-STING-IRF3-IFN-I pathway activation. We demonstrated that p-STING and p-IRF3 expression was upregulated at 16 h post HSV-1 infection, but FUT8 overexpression or knockdown did not affect cGAS, p-STING and p-IRF3 expression (Supplementary Fig. 5d),

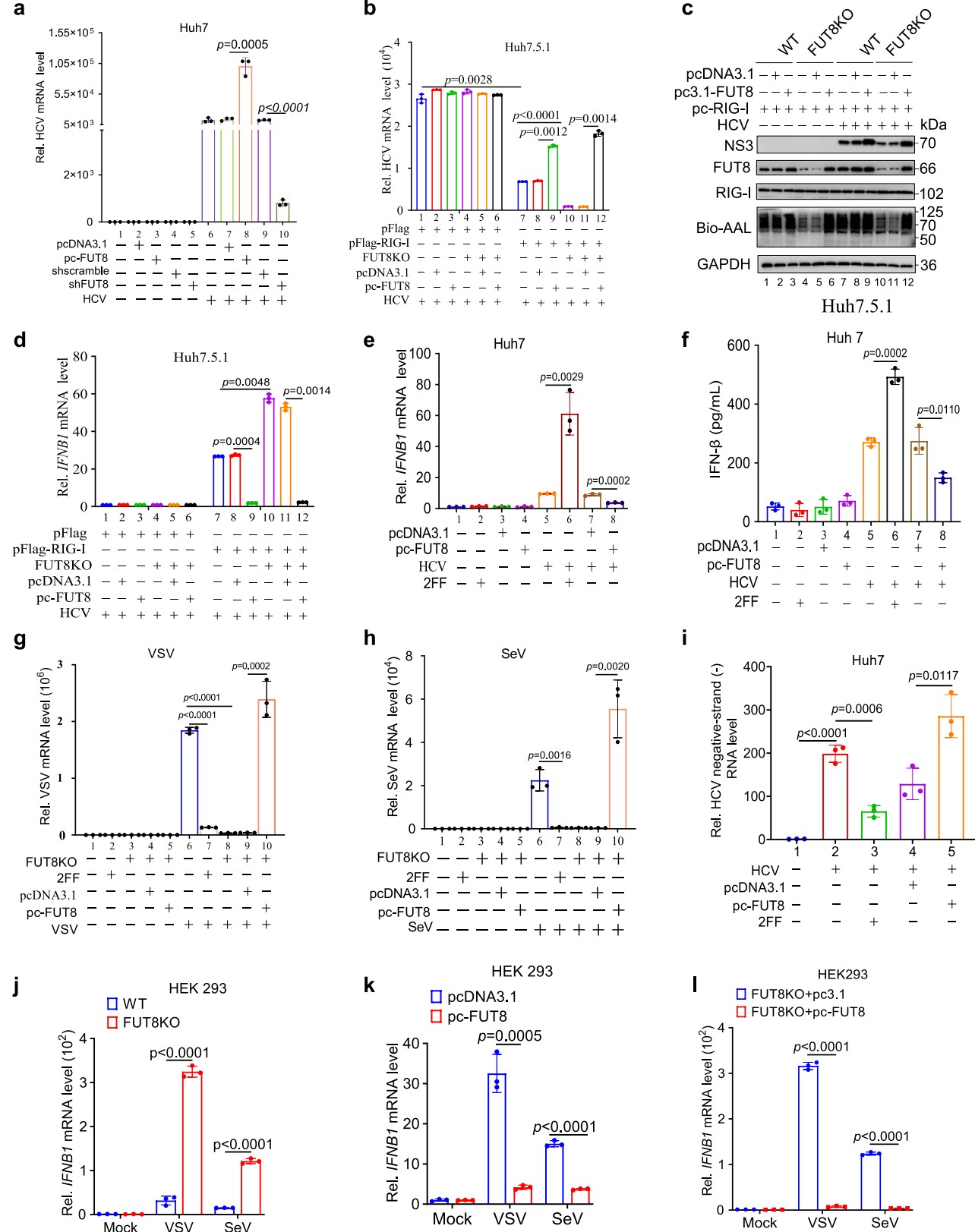

thus did not affect HSV-1-induced IFN-β expression (Supplementary Fig. 5b) and HSV-1 viral replication (Supplementary Fig. 5c).

Above findings demonstrate that upregulation of FUT8 inhibits cellular RIG-I-induced type I IFN production and thus increases RNA viral replication but has no effect on DNA viral replication.

## Hepatic targeted *Fut8* silencing and FUT8 inhibitor 2FF suppress RNA viral replication in vivo

To confirm the effects of FUT8 on viral replication in vivo, we established HCV-infected mouse models. We used *Fut8* tissue-specific knockdown in HCV-infected ICR[4R+] transgenic mouse model or the

**Fig. 5 | FUT8 promotes RNA viral replication and suppresses RIG-I-induced type I IFN production. a** RT-qPCR analysis of HCV mRNA expression in Huh7 cells transiently transfected with indicated plasmids for 48 h, and then infected with HCV (MOI = 0.01) for 48 h (*n* = 3 per group per study). RT-qPCR analysis of HCV mRNA expression (**b**) and immunoblot analysis of HCV NS3 (**c**) in WT or FUT8KO Huh7.5.1 cells transiently co-transfected with indicated plasmids for 48 h, and then infected HCV (MOI = 0.01) for 48 h (**b**, *n* = 3 per group per study) or 72 h (**c**). RT-qPCR analysis of *IFNB1* mRNA expression in Huh7.5.1 cells (**d**) or Huh7 cells (**e**) or ELISA of IFN-β in supernatants of Huh7 cells (**f**) transiently transfected with indicated plasmids or treated with 2FF for 48 h, and then infected with HCV (MOI = 0.01) for 12 h or 48 h (**d–f**, *n* = 3 per group per study). RT-qPCR analysis of virus mRNA expression in FUT8KO HEK293 cells transiently transfected with indicated

plasmids or WT HEK293 cells treated with 2FF for 48 h, and then infected with VSV (**g**) (MOI = 0.001) or SeV (**h**) (MOI = 0.01) for 48 h (**g**, **h**, *n* = 3 per group per study). **i** Strand-specific Tth-based RT-qPCR analysis of HCV-negative strand RNA in Huh7 cells transiently transfected with pcDNA3.1-FUT8 or treated with 100 μM 2FF for 48 h, and then infected with HCV (MOI = 0.01) for 48 h (*n* = 3 per group per study). RT-qPCR analysis of *IFNB1* mRNA expression in WT (**k** and **l**) or FUT8KO (**j** and **l**) HEK293 transiently transfected with indicated plasmids or not for 48 h, and then infected with VSV (MOI = 0.001) or SeV (MOI = 0.01) for 12 h (**j–l**, *n* = 3 per group per study). Data are normalized based on *GAPDH* for **a**, **b**, and **d–l**. Data in all quantitative panels are presented as mean ± SD. Data are representative of three independent experiments. Two-tailed unpaired Student's *t* test was used to assess the statistical difference in **a**, **b**, **d–l**. Source data are provided as a Source Data file.

FUT8 inhibitor 2FF in VSV-infected C57BL/6J mouse infection model in the following experiments.

We performed hepatic-targeted *Fut8* silencing using a transposase in ICR[4R+] humanized mice[59,60]. We constructed the plasmid pT3-U6-shmFut8, which was equipped with an inverted terminal repeat sequence, loxP, at both ends of the U6-Fut8 shRNA. This sequence can bind to a transposase and be inserted into the target cell genome[61]. Hepatic-targeted *Fut8* silencing was achieved through hydrodynamic tail vein injection of each of the above plasmids with Sleeping Beauty (SB100)-mediated somatic cell integration[62–64], which drives the plasmids into the hepatic vein and hepatocytes owing to the increase in intravascular pressure in the inferior vena cava upon tail vein injection. After 7 d post injection of plasmids (pT3-U6-shRNA/pT3-U6-shScramble and SB100), we intravenously (*i.v.*) injected HCVcc (1 ×10⁶ copies) into each ICR[4R+] humanized mouse. The mice were humanely sacrificed, and their livers, spleens, lungs and kidneys were collected 30 d post-injection (Fig. 6a). Hepatocyte-specific knockdown of *Fut8* led to decreased expression of FUT8 in the mouse liver tissues, but not in other organs such as spleens, lungs and kidneys as shown using WB (Fig. 6b and Supplementary Fig. 6a). Viral proteins NS3 and Core expression and viral RNA levels decreased by *Fut8* hepatic-specific knockdown by WB and RT-qPCR (Fig. 6b, c, pT3-U6-sh*Fut8 vs.* pT3-U6-shScramble). *Fut8* hepatic-targeted knockdown increased IFN-I expression in the mouse sera as shown using the enzyme-linked immunosorbent assay (ELISA) (Fig. 6d) and in liver by RT-qPCR (Fig. 6e). Immunohistochemical results showed that after knocking down *Fut8* in the liver of ICR[4R+] mice, the HCV Core protein expression was inhibited, compared with shScramble/PBS + HCV groups (Fig. 6f, lower row *vs.* the two medium rows).

Liver inflammatory infiltration is a histological feature, usually representing that immune cells have been recruited to the liver during HCV infection[65,66]. The presence and nature of portal inflammatory infiltrates can help assess the extent of liver damage and inflammation caused by the virus. The boundary between the white pulp and red pulp in the spleen can become blurred, and lymphoid tissue in the white pulp may undergo hyperplasia (an increase in cell numbers) appearing as large masses within the spleen, due to the inflammation and damage[67,68]. Histopathological examination of the tissues (Supplementary Fig. 6b, c) revealed the following: (a) Portal inflammatory infiltrates (inside dashed white circles) were observed in PBS plus HCV group and shScramble plus HCV group, but fewer in shFut8 plus HCV group and none in only PBS group (Supplementary Fig. 6b). (b) As a result of HCV infection, the white pulps joined to form a large mass with blurred boundary between white pulp and red pulp in PBS plus HCV group and shScramble plus HCV group, but these changes were less pronounced in shFut8 plus HCV group and none in PBS group (Supplementary Fig. 6c). These data strongly suggest that hepatic-targeted *Fut8* silencing suppresses HCV RNA replication, and alleviates inflammation and tissue damage in vivo.

In addition, we detected the similar inhibitory effects of FUT8 on VSV replication in C57BL/6J mice using the FUT8 inhibitor 2FF. Mice were tail vein-injected with 2FF (5 mg/kg) consecutively for 7 d,

followed by intranasal infection with VSV. After 48 h post-infection (Supplementary Fig. 7a), the 2FF-treated mice showed suppressed lung VSV replication (Supplementary Fig. 7b) and VSV-G protein expression (Supplementary Fig. 7c). AAL blotting assay also showed that core fucosylation decreased after 2FF treatment (Supplementary Fig. 7c, lane 8–10). We also observed enhancement of *Ifnb1* (Supplementary Fig. 7d) and IFN-stimulated genes (ISGs; MX dynamin-like GTPase 1 [*Mx1*] (Supplementary Fig. 7e), 2'-5'-oligoadenylate synthetase 2 [*Oas2*] (Supplementary Fig. 7f) and chemokine [C-X-C motif] ligand 10 [*Cxcl10*] (Supplementary Fig. 7g) mRNA expression. Lung histopathology showed that the VSV infection group (Supplementary Fig. 7h, medium panel) had severe lung perivascular infiltration and alveolar septal thickening compared to the PBS-treated uninfected group (Supplementary Fig. 7h, medium panel *vs.* upper panel). The 2FF-treated group showed less perivascular lung infiltration and alveolar septal thickening than the VSV-infected PBS-treated group (Supplementary Fig. 7h, lower panel *vs.* medium panel). These data strongly suggest that FUT8 inhibitor suppresses VSV RNA replication, upregulates IFN-I and ISGs expression, and alleviates tissue damage in vivo.

## Suppression of FUT8 upregulates anti-viral innate immune response genes

Interestingly, based on microarray analysis of data downloaded from the Gene Expression Omnibus (GEO) database (GSE42405), we found that differentially expressed genes (DEGs) in the top ten enriched biological processes were associated with innate and adaptive immune responses in FUT8 knockdown cells compared with those in WT control cells (Supplementary Fig. 8a–c). Most of the upregulated genes in FUT8 knockdown cells were associated with response to external stimuli (CDKN1A, ATF2, DNAJC15, IFI16, MN1, PDE2A, SFRP1, and TLR4), regulation of viral processes (IFI16, ISG20, OAS1, OAS2, and TRIM8), viral genome replication (IFI16, ISG20, OAS1, and OAS2), and response to viruses (IFI44, ATF2, IFI16, IFIT2, ISG20, OAS1, OAS2, and TRIM8) (Supplementary Fig. 8a–c). Notably, our RT-qPCR results confirmed that a panel of genes responding to external stimuli and viruses were significantly upregulated in FUT8 KO HEK293 (Supplementary Fig. 8d) or 2FF-treated Huh7 cells (Supplementary Fig. 8e), and were obviously downregulated in FUT8 KO plus FUT8 complemented HEK293 (Supplementary Fig. 8f) or FUT8 overexpression Huh7 cells (Supplementary Fig. 8g). These results suggest that FUT8 inhibition can upregulate a panel of immune response genes to trigger antiviral immunity.

## Discussion

Up till now, only a few reports have focused on the modification of the host cell glycosylation profile and activation of host cell glycosyltransferase transcription during viral infection, although extensive studies have investigated the impact of viral glycosylation modification on viral life cycles[69–73]. In the present study, we found that multiple viral envelope proteins (HCV-E2, VSV-G, HIV-gp120 and SARS-CoV-2-Spike) upregulated host cellular FUT8 glycosyltransferase expression, but SeV or bacterial *E. coli* infection had no such effect. The main

**a**

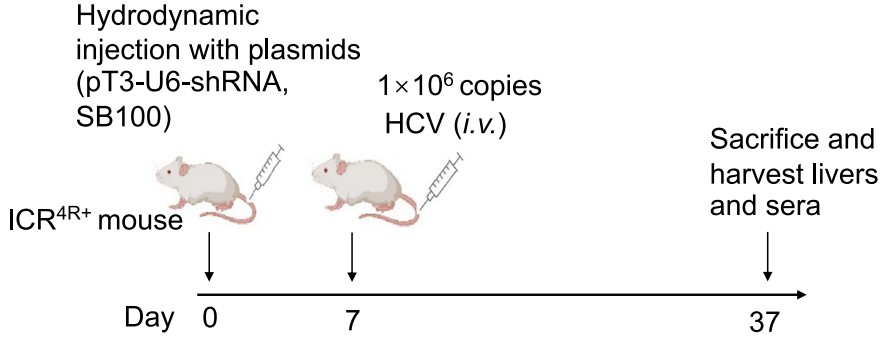

**b**

pT3-U6-shScramble − − + −
pT3-U6-shFut8 − − − +
HCV − + + + kDa

Bio-AAL — 150 / 70 / 50

FUT8 — 66

NS3 — 70

Core — 20

GAPDH — 36

**c**

Rel. HCV mRNA level ($10^3$)

p=0.0038
p=0.0102

pT3-U6-shScramble − − + −
pT3-U6-shFut8 − − − +
HCV − + + +

**d**

IFN-β (pg/mL)

p<0.0001
p=0.0008

pT3-U6-shScramble − − + −
pT3-U6-shFut8 − − − +
HCV − + + +

**e**

Rel. *Ifnb1* mRNA level

p=0.0002
p=0.0004

pT3-U6-shScramble − − + −
pT3-U6-shFut8 − − − +
HCV − + + +

**f**

anti-FUT8    anti-Core

PBS

PBS+HCV

pT3-U6-shScramble +HCV

pT3-U6-shFut8 +HCV

100 μm

ICR^4R+ mouse liver tissues

**Fig. 6 | FUT8 silencing suppresses HCV RNA replication and viral protein translation in ICR^4R+ transgenic mouse infection models. a** Scheme of mouse HCV infection models. The figure is created with Biorender.com. **b** Lectin and immunoblot analysis of core fucosylation, FUT8, NS3 or Core in the liver tissues of ICR^4R+ mice infected with HCV for 30 d. **c** RT-qPCR analysis of HCV RNA in the liver tissues of ICR^4R+ mice infected with HCV for 30 d (*n* = 3 mice per group per study). **d** ELISA of serum IFN-β in ICR^4R+ mice infected with HCV for 30 d (*n* = 3 mice per group per study). **e** RT-qPCR analysis of *Ifnb1* in the liver tissues of ICR^4R+ mice infected with HCV for 30 d (*n* = 3 mice per group per study). **f** Immunohistochemical staining for FUT8, HCV Core protein in liver tissue of HCV-infected ICR^4R+ mice. Data in all quantitative panels are presented as mean ± SD. Data are representative of three independent experiments. Data are normalized based on *Gapdh* for **c** and **e**. Two-tailed unpaired Student's *t* test was used to assess the statistical difference in **c**–**e**. Source data are provided as a Source Data file.

difference between enveloped and non-enveloped viruses is the outer protective envelope glycoproteins that cover the viruses. SeV belongs to the Paramyxoviridae family and is in fact enveloped. We observed that SeV could not activate p-EGFR, thus could not induce p-EGFR-SNAIL-FUT8 axis activation and core fucosylation (Supplementary Fig. 3h). Lupberger et al. also reported that measles virus, a related Paramyxoviridae family member, does not require EGFR for infection[74]. Although these enveloped viruses, except SeV, induced FUT8, the extent of FUT8 induction varies. For example, at the same MOI (0.1), VSV induced about 25-fold (Fig. 1h), while HCV induced about 12-fold FUT8 upregulation (Fig. 1a). Different viral envelope proteins might engage different receptors, and they might co-opt receptors in different degrees.

We also found that N-glycosylation site-specific mutation (Q352AT) of EGFR inhibited the ability of EGF to stimulate EGFR endocytosis (Fig. 3d and Supplementary Fig. 4f). Envelope viruses induce EGFR endocytosis by interacting with EGFR, we speculate that envelope viruses preferentially interact with EGFR. Consistently, multiple enveloped viruses (including HCV[74], influenza virus IAV[23], and human cytomegalovirus HCMV[75]) interact with EGFR to facilitate viral entry and replication. The binding of HCV particles and HCV-E2 protein to human hepatocytes induces EGFR activation[45]. Many viral infections (e.g., HCV, VSV, and HSV-1) stimulate the EGFR/AKT endocytosis signaling pathways[40–43].

Which host transcription factors are manipulated by viruses to induce FUT8 expression remains to be explored. In the present study, we found that HCV infection and its envelope protein E2 upregulated FUT8 through EGFR-AKT-SNAIL activation and nuclear translocation of the SNAIL transcription factor. To the best of our knowledge, this is the first report on SNAIL regulation of FUT8-mediated core fucosylation during viral infection. SNAIL may be an attractive target for the modification of core fucosylation. It would be interesting to examine whether SNAIL activation is correlated with other glycosyltransferases involved in viral infections.

Core fucosylation is essential for EGFR-mediated biological functions in tumors[10,24,76–78]. FUT8 triggered the core-fucosylated-Hsp90/MUC1/P300-HOTAIR-STAT3 cascade through the JAK1/STAT3 pathway, which exhibits a positive feedback loop during HCC progression[79]. However, the role of EGFR glycosylation and core fucosylation in viral infections remains unclear. Our present data demonstrated that FUT8-induced core fucosylation of EGFR promoted RNA viral replication. The EGFR N-glycosylation site at N352 is critical for EGFR-mediated enhancement of HCV RNA replication. We found that 14 glycosylation site mutations in EGFR did not affect EGFR protein expression (Fig. 3d), indicating that glycosylation site-specific mutations caused by gene engineering are less toxic to cells and do not affect protein synthesis. These findings suggest that modification of glycosylation and core fucosylation of EGFR may be a potential strategy for controlling HCV infection and viral RNA replication.

Our data demonstrated that FUT8 decreased the production of IFN-I by promoting RIG-I K48-linked ubiquitination and proteasomal degradation, which increased the replication of RNA viruses, but not DNA viruses. RIG-I, an intracellular viral RNA sensor, initiates an anti-RNA viral IFN-I innate immune response[80]. cGAS-STING is an intracellular viral DNA sensor that initiates anti-DNA viral IFN-I innate immune responses[81]. Ubiquitination is one of the most versatile posttranslational modifications that is indispensable for cellular homeostasis, including innate antiviral immune responses. Posttranslational modification of RIG-I and downstream signaling proteins by different types of ubiquitination have been found to be key events in the regulation of RIG-I-induced NF-κB and IRF3 activation[82]. In line with previous reports[83], we found that FUT8-induced p-STAT3 expression mediated RIG-I downregulation. However, the specific E3 ligase (TRIM40)[53] that mediates polyubiquitination and degradation of RIG-I by STAT3 remains unknown. In the present study, we demonstrated that

FUT8-induced pSTAT3 recruiting TRIM40 to mediate polyubiquitination and degradation of RIG-I. Thus, further investigation is required to identify the ubiquitination sites and ubiquitin ligases or proteins mediated by STAT3 that regulate RIG-I signaling. Our findings provide new insights into the biological functions of FUT8 in RIG-I-mediated anti-viral innate immunity. There are possibly other potential mechanisms that affect RIG-I activation (beyond its degradation). Other studies have shown that deubiquitinases (such as USP21), LGP2, and PKCα/β could suppress RIG-I activation[84]. Whether FUT8 regulates these pathways to induce RIG-I degradation needs further investigation.

FUT8 inhibitor 2FF inhibits cellular protein fucosylation. 2FF has been experimentally applied in tumor therapy[85]. Administration of 2FF as prophylaxis significantly delayed tumor onset and improved overall survival[86]. Inhibition of FUT8 by 2FF in human invasive ductal carcinoma reduced E-selectin ligand expression, cell proliferation, and ERK1/2 and p38 MAPK activation[87]. 2FF application in virus prevention has rarely been reported. Here, we found that the FUT8 inhibitor, 2FF, suppressed RNA viral replication (including HCV and VSV) in both cell culture and mouse infection models. We demonstrated here that hepatic targeted Fut8 silencing/FUT8 inhibitor 2FF suppressed HCV/VSV RNA viral replication in vivo. Our results confirmed that the FUT8 inhibitor could be used as an efficient antiviral drug. However, more work is needed to develop this therapy, such as the design of small-molecule compounds with high efficiency, low toxicity, and high specificity for tissue-targeted carriers.

The limitations of the present study are as follows: (a) although we found HCV E2, VSV-G, SARS-CoV-2-spike and HIV-gp120 envelope proteins preferably induce EGFR endocytosis and virus-induced FUT8 caused EGFR core fucosylation, FUT8 might also induce core fucosylation of other cellular membrane proteins. The detailed mechanisms involved in virus infection still require further investigation. (b) due to the insufficiency and sophistication of the infection mouse models for other viruses, we only demonstrated that FUT8 promoted RNA viral replication (HCV and VSV) in both cellular and mouse infection models. Future studies for other viruses are needed.

In summary, we discovered that FUT8 is upregulated by HCV infection through EGFR-AKT-SNAIL activation, and then FUT8 regulates RIG-I signaling-mediated anti-RNA viral innate immune responses through a negative feedback mechanism. FUT8 induces EGFR core fucosylation, which further promotes the activation of EGFR-JAK1-STAT3 signaling, K48-linked ubiquitination and proteasomal degradation of RIG-I (Fig. 7). These findings provide insights into the mechanism of the immunological escape of RNA viruses and potential therapeutic targets for RNA virus-associated diseases. FUT8 might be a promising target for viral therapies.

## Methods

### Cell culture and virus infection
Human hepatocellular liver carcinoma (Huh7.5.1 and Huh7)[88,89] and human embryonic kidney (HEK 293) cells were grown in Dulbecco's Modified Eagle's Medium (DMEM; Gibco, USA, Cat# 6123066) with 10% fetal bovine serum (FBS; Gibco, USA, Cat# 2010162 C) at 37 °C with 5% CO2. The JFH1 (Japanese fulminant hepatitis 1, genotype 2a) HCV cell culture (HCVcc) was performed as previously described[90–92]. In brief, Huh7 cells in 24-well plates were infected with HCVcc (MOI = 0.1) at 37 °C for 4 h. The supernatants were discarded, and the infected cells were washed twice with PBS and incubated in DMEM containing 10% FBS for each experiment. FUT8KO Huh7.5.1 cells were generated using Genloci with CRISPR/Cas9. FUT8KO HEK 293 cells were provided by Dr. Xiao-Dong Gao of Jiangnan University. EGFR KO Huh7 cells were generated by CRISPR/Cas9 using the Lentiviral CRISPR gRNA Sanger with the target site TAACCAGCCACCTCCTGGATGG. Vesicular stomatitis virus (VSV, a prototypical member of the Rhabdoviridae family with a single-stranded, negative-sense RNA), Sendai virus (SeV, a

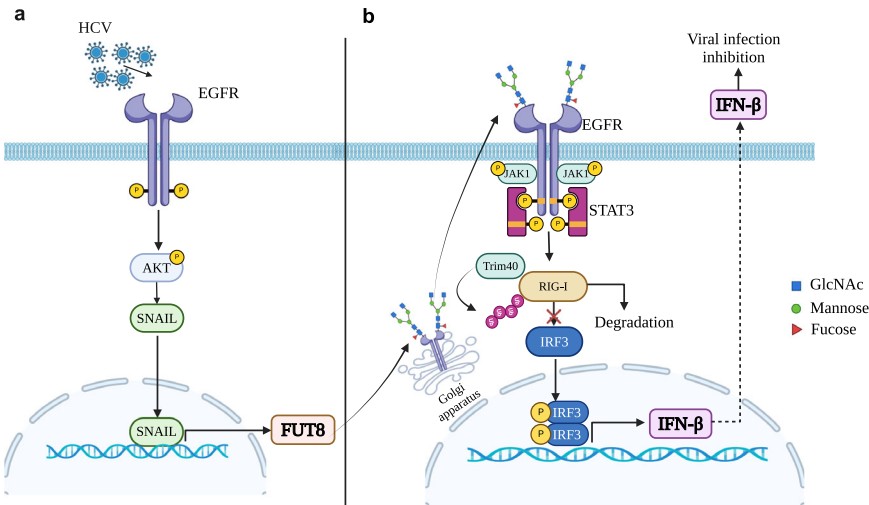

**Fig. 7 | Diagram showing that HCV upregulated-FUT8 promotes core fucosylated-EGFR-STAT3 and RIG-I degradation.** HCV infection upregulates FUT8 protein expression through EGFR-AKT-SNAIL axis (**a**), and promotes EGFR core fucosylation and EGFR/STAT3 phosphorylation (**b**). The aberrant increased-FUT8 expression promotes Trim40 to associate with RIG-I, leading to K48-linked ubiquitination and proteasomal degradation of RIG-I, which suppresses RIG-I-triggered IFN-I production in the innate immune response. The figure is created with Biorender.com.

paramyxovirus with a single-stranded, negative-sense RNA), and herpes simplex virus (HSV, a prototypic herpes virus, and double-stranded DNA virus) were kindly provided by Dr. Fei Deng of the Chinese Academy of Sciences. Hepatitis C virus (HCV, a positive-sense single-stranded RNA virus) and *E. coli* DH5α are maintained in our lab. The SARS-CoV-2 pseudovirus was provided by Prof. Huan Yan of Wuhan University (Wuhan, China). The HIV-env plasmid (pcDNA3.1D/V5-His-TOPO) was kindly provided by Dr. BinLian Sun of Wuhan Institute of Virology, Chinese Academy of Sciences (Wuhan, China)[93,94]. For assessing the effects of viruses or bacteria on FUT8 expression, in brief, HEK293/Hela cells in 24-well plates were infected with VSV/SeV/HSV-1 (MOI = 0.1) or SARS-CoV-2 pseudovirus (MOI = 2) at 37 °C for 4 h or *E. coli* DH5α (bacterium: cell = 100:1) at 37 °C for 1 h. The supernatants were discarded, and the infected cells were washed twice with PBS and incubated in DMEM containing 10% FBS for each experiment.

### Reagents and antibodies
The FUT8 inhibitor 2FF and the EGFR phosphorylation inhibitor erlotinib were purchased from Topscience (Topscience, China, Cat# T12894). EGF was purchased from PeproTech (Rockville, MD, Cat# AF-100-15). Bio-AAL was purchased from Vector Laboratories (Burlingame, CA, Cat# B-1395). Biotin-AOL was kindly provided by Prof. Wenzhe Li of Dalian Medical University, China (TCI, Japan, Cat# A2659). Anti-His-tag (Cat# 66005-1-Ig) and anti-Trim40 (Cat# 67073-1-Ig) were purchased from Proteintech Group, Inc. (Rosemont, IL, USA). Polyclonal anti-FUT8 (Cat# ab198741), monoclonal anti-NS3 (Cat# ab13830), anti-core (Cat# ab2740), anti-HIV1 gp120 (Cat# ab85054), anti-Ubiquitin (linkage-specific K48) (Cat# ab140601), anti-RIG-I (Cat# ab45428) and anti-VSV-G (Cat# ab309106) were from Abcam (Cambridge, UK). The anti-AKT (Cat# 4691S) and anti-pAKT (Cat# 4060S) antibody were purchased from Cell Signaling Technology (Danvers, MA, USA). Anti-GAPDH (Cat# AC002), anti-EGFR (Cat# A11351), anti-p-EGFR-Y1068 (Cat# AP0301), anti-SNAIL (Cat# A5243), anti-JAK1 (Cat# A18323), anti-p-JAK1 (Cat# AP0530), anti-STAT3 (Cat# A11216), anti-p-STAT3 (Cat# AP0705), anti-IRF3 (Cat# A0816), anti-p-IRF3 (Cat# AP0623), anti-SARS-CoV-2 Spike S (Cat# A20136), anti-CD81 (Cat# A5270), anti-OCLN (Cat# A2601), anti-CLDN-1 (Cat# A2196) and anti-SR-B1 (Cat# A0827) antibodies were purchased from ABclonal (Wuhan, China). Anti-cGAS (Cat# PAB47500), anti-STING (Cat# RMAB50052), and anti-p-STING (Cat# PAB47864-P) were purchased from Bioswamp (Wuhan, China). Alanine Aminotransferase (ALT/GPT)

activity fluorometric assay kit was obtained from Elabscience Biotechnology Co., Ltd., China, Cat# E-BC-F038.

Antibodies for immunoblot were used at a dilution of 1:1000 - 1:2000. Antibodies for immunoprecipitation were used at a dilution of 1:100 - 1:200. AAL for lectin blot was used at a dilution of 1:2000. AOL for lectin blot was used at a dilution of 1:5000.

### Plasmids and transfection
Myc/his-tagged expression plasmids for the HCV core, P7, E1, E2, NS2, NS3, NS4a, NS4b, NS5a, and NS5b proteins were cloned into the pcDNA3.1-Myc-His expression vector using standard cloning techniques. FLAG-tagged wild-type or each N-glycosylation site-specific mutation of EGFR variants were generated using the 2X MultiF Seamless Assembly Mix kit (ABclonal, China, Cat# RK21020). Expression vectors for all EGFR N-glycosylation mutants were verified by DNA sequencing. Each EGFR mutant had a flag-tagged epitope at its N-terminus, and all primers are listed in Supplementary Table 1.

Genomic DNA was extracted from Huh7 cell lysates using a genomic DNA extraction kit (TIANGEN, China, Cat# DP304). Primers were designed based on the sequence of the 5′ untranslated region of human FUT8 (hFUT8) (GenBank accession No. BK008802). Isolated genomic DNA was used as a template to amplify the hFUT8 promoter region. *Kpn*I and *Hin*dIII restriction sites were introduced into forward and reverse primers, respectively. The PCR product was digested with *Kpn*I and *Hin*dIII, and then ligated into pGL3-Basic, a plasmid containing firefly luciferase with no promoter.

To generate vectors encoding shRNAs that target specific genes (FUT8, EGFR, JAK1, AKT, SNAIL, and STAT3), oligo pairs harboring sense and antisense sequences were synthesized, annealed, and cloned into the *Age*I and *Eco*RI-digested pLKO.1 vector (Addgene plasmid Cat# 10879). The shRNA sense and antisense sequences were designed and constructed based on the siRNA Selection Program (https://rnaidesigner.thermofisher.com/rnaiexpress/) and are listed in Supplementary Table 2. Silencing efficiencies were verified using WB.

Cells were grown to 50% confluence in 6-well plates and transfection was performed using Lipofectamine 2000 (Invitrogen, Carlsbad, CA, USA, Cat# 11668030), according to the manufacturer's instructions. In all the co-transfection experiments, the corresponding empty vectors were used as negative controls to ensure similar DNA concentrations. Transfection efficiency was verified using RT-qPCR or WB.

## RNA extraction and real-time RT-qPCR

Total RNA was extracted using the TRIzol reagent (Invitrogen, USA, Cat# 15596018). Reverse transcription (Toyobo, Japan, Cat# FSQ-101) was performed using 500 ng purified RNA as a template. The obtained cDNA samples were subjected to PCR using a PCR Kit (Toyobo, Japan, Cat# QPK-201). RT-qPCR was performed on an ABI StepOnePlus instrument (Applied Biosystems) under standard cycling conditions. The relative mRNA expression level of each gene was normalized to that of GAPDH. Quantification of transcriptional level was calculated using the $2^{-\Delta\Delta Ct}$ method. All experiments were performed at least thrice. All primers used for RT-qPCR are listed in Supplementary Table 3.

## Immunoblotting (IB), immunoprecipitation (IP) and lectin blotting

For IB and lectin blotting, whole-cell lysates in RIPA buffer were supplemented with a protease inhibitor cocktail for 30 min on ice. Cell lysates were separated using SDS-polyacrylamide gel electrophoresis (PAGE) and transferred onto polyvinylidene fluoride (PVDF) membranes by electroblotting. After blocking with 5% BSA in TBS for IB or PBS containing 0.05% Tween 20 (TBST or PBST) for lectin blotting, the membrane was incubated with the antibody of interest and then with peroxidase-conjugated anti-rabbit IgG. GAPDH was used as the control. All band intensities were evaluated using an ECL WB kit (HYCEZMBIO, China, Cat# HYC0316) and normalized to those of GAPDH. Three independent experiments were performed for each analysis.

For IP, the cells were lysed in RIPA buffer supplemented with a protease inhibitor cocktail on ice for 30 min. Primary antibodies were incubated with protein A agarose beads (MCE, USA, Cat# HY-K0213) at 4 °C for 4 h, followed by incubation with incubation with cell lysates for 5 h with rotation at 4 °C. Beads were washed four times with lysis buffer and analyzed using IB.

## Luciferase reporter assay

The cells were transfected with the reporter plasmid pGL3 containing the IFN-β promoter and the indicated amounts of expression plasmids. The pRL-TK Renilla luciferase reporter plasmid was added to each transfection reaction to normalize the transfection efficiency. Luciferase reporter assays were performed using a dual-luciferase reporter assay system (Promega, USA, Cat# E1910) according to the procedures provided by the manufacturer's instructions. Firefly luciferase activity was normalized to the Renilla luciferase activity. Relative luciferase activities were expressed as fold-changes over the empty-plasmid-transfected or mock controls.

## Recombinant protein expression, purification and EMSA

The His6-SNAIL expression vector was constructed using pET28a and transformed into *E. coli* strain Rosetta (DE3). His6-SNAIL protein expression was induced using 1 mM isopropyl 1-thio-β-D-galactopyranoside (IPTG) at 16 °C for 13 h, and the recombinant His-fusion protein was eluted using buffer containing 200 mM imidazole.

The fragment of approximately 30 bp containing putative SNAIL binding sequence in the promoter region (−295 to −260 bp) of FUT8 or its mutant was labeled with FAM at the 5' end. An EMSA was performed using the Chemiluminescent EMSA Kit (Beyotime, China, Cat# GS005) according to the manufacturer's protocol. The purified His6-SNAIL fusion protein was incubated with Fam-labeled DNA fragments, and the protein-DNA complexes were separated using 8% PAGE under native conditions, followed by detection on a Typhoon™ FLA 9500 biomolecular imager (GE Healthcare, Piscataway, NJ, USA) using chemiluminescence. A 10-fold molar excess of unlabeled DNA fragments with the same or mutant sequences was used as competitors. The primers used for the EMSA are shown in Supplementary Table 4.

## Bioinformatics analysis of microarray data

Publicly available data were downloaded from the GEO database (GSE42405) and then standardized and calibrated, and the log-expression matrix was extracted from MAList with R package limma (3.52.2 version). The DEGs were identified with an adjusted $p$-value < 0.05, absolute log2 fold-change >1, and plotted with the R package ggplot2 (v3.3.6). Gene Ontology (GO) enrichment analysis of the upregulated DEGs was performed using the R package clusterProfiler (v4.4.4). Heat maps were generated using the pheatmap package (v1.0.12). The upregulated genes and pathways in FUT8 knockdown cells are shown in Supplementary Table 5.

## Measurement of IFN-β cytokine level using ELISA

Huh7 cells ($10^5$ cells in 12-well plates) were transiently transfected with His-vector or His-FUT8 plasmids or treated with 2FF (100 μM) for 48 h, and then infected with HCV for 48 h. The supernatants were collected for IFN-β detection using ELISA. The IFN-β cytokine ELISA was performed according to the manufacturer's instructions (Bioswamp, Wuhan, China, Cat# HM10099), and the OD value was read at 450 nm. The assays were repeated at least thrice.

## Ubiquitination assay

The HA-tagged ubiquitin plasmids were maintained in the laboratory[95]. HEK293T cells were transfected with plasmids encoding His-RIG-I, HA-ubiquitin, and Flag-STAT3 for 24 h and then infected with HCV for 48 h. Half of each cell aliquot was treated with 10 mM MG132, a proteasome inhibitor that blocks the degradation of ubiquitin-conjugated RIG-I. Cells were harvested for immunoblot analysis of Ub immunoprecipitated using an antibody against the tag. Ubiquitinated RIG-I was detected using IB with a specific K48-ubiquitin antibody.

## Animal infection experiment

All animal experiments were approved by the Ethics Committee of Center for Animal Experiments at Wuhan University (No. AF060, S01319070T). All mice were kept under specific-pathogen free conditions in Animal Facility of Center for Animal Experiments at Wuhan University. These mice were kept in an animal room with a 12-h light-dark cycle at a temperature of 20–25 °C with 40–60% humidity. Six-week-old male C57BL/6J mice were treated with 2FF (5 mg/kg) consecutively for 7 d followed by intranasal infection with 200 μL of VSV (2 ×$10^6$ PFU per mouse). Three to four mice were used for each group. Mice were sacrificed 48 h post-infection, lungs were collected, and viral infection were analyzed using WB, RT-qPCR, and H&E staining.

A humanized murine model of persistent HCV infection, ICR$^{4R+}$ (expressing four human HCV receptors: CD81, OCLN, SR-B1, and CLDN1), was constructed on an ICR genetic background (Beijing Vitalstar Biotechnology Co., Ltd., Beijing, China)[96]. Eight-week-old male mice were injected with JFH-1 HCVcc (4 ×$10^6$ plaque-forming units [PFU] per mouse). On day 40 post-infection, the mice were sacrificed, liver tissues were collected, and the mRNA expression levels of FUT family genes were detected using RT-qPCR.

For hepatic targeted *Fut8* silencing experiment, ICR$^{4R+}$ mice were used and the hydrodynamic injection procedures were performed as previously described[97]. Briefly, 10 μg of the pCMV/SB and pT3-U6-shFUT8 constructs at a ratio of 1:25 were diluted in 2 mL of PBS, filtered, and injected into the lateral tail vein of 6-week-old male mice for 7–9 s. After 1 week, the mice were challenged with HCVcc (1 ×$10^6$ copies). Mice were sacrificed at Week 4, and hepatocytes were obtained for WB and RT-qPCR.

## Mouse liver function test

Mouse sera were collected for biochemical assays. Briefly, serum alanine aminotransferase (ALT) was quantified using ALT/GPT Activity Fluorometric Assay Kit (Elabscience, Wuhan, China, Cat# E-BC-F038).

## Preparation of nuclear and cytoplasmic extract

Nuclear and cytoplasmic proteins were extracted using a Nuclear and Cytoplasmic Protein Extraction Kit (Beyotime Institute of Biotechnology, Jiangsu, China, Cat# P0027), according to the manufacturer's instructions.

## Immunofluorescence microscopy analysis

To analyze the colocalization of EGFR and LAMP1, Huh7 cells ($3 \times 10^5$) were seeded in confocal dishes (NEST Biological Technology Co., Ltd., Shanghai, China) and transfected with combinations of eukaryotic plasmids (pFlag-EGFR, a plasmid encoding Flag-EGFR). After transfection for 48 h, the cells were incubated on ice with 10 nM (EGF), washed, and incubated at 37 °C for 60 min. For immunofluorescence microscopy analysis, cells were fixed with 4% paraformaldehyde for 15 min at 25 °C and permeabilized for 10 min with 0.1% Triton X-100. Samples were blocked with 5% BSA in phosphate-buffered saline (PBS) at 37 °C for 30 min. EGFR polyclonal antibody (dilution 1:200, ABclonal, China, Cat# A11351) and CoraLite®647 Anti-human LAMP1 (dilution 1:400, Proteintech, USA, Cat# Cl647-65051) were used at 4 °C for 12 h to detect the EGFR protein and lysosomal, respectively. Then, samples were incubated with anti-rabbit IgG (H + L), F(ab')$_2$ Fragment (Alexa Fluor® 488 Conjugate) (Cell Signaling Technology, USA, Cat# 4412) at 25 °C for 1 h. The cell nuclei DNA was stained with DAPI. The expression and colocalization of EGFR and lysosomal were analyzed using a confocal laser scanning microscope (Zeiss LSM 880 objective 60×oil).

To analyze the colocalization of EGFR, FUT8 and GM130, Huh7 cells ($3 \times 10^5$) were seeded in confocal dishes (NEST Biological Technology Co., Ltd., Shanghai, China) and infected with HCV for 48 h. For immunofluorescence microscopy analysis, cells were fixed with 4% paraformaldehyde for 15 min at 25 °C and permeabilized for 10 min with 0.1% Triton X-100. Samples were blocked with 5% BSA in phosphate-buffered saline (PBS) at 37 °C for 30 min. EGFR polyclonal antibody (dilution 1:200, ABclonal, China, Cat# A11351) and FUT8 monoclonal antibody (dilution 1:100, Proteintech, USA, Cat# 66118-1-Ig) were used at 4 °C for 12 h to detect the EGFR and FUT8, respectively. Samples were incubated with anti-rabbit IgG (H + L), F(ab')2 Fragment (Alexa Fluor® 488 Conjugate) and anti-mouse IgG (H + L), F(ab')2 Fragment (Alexa Fluor® 647 Conjugate) at 4 °C for 12 h. Then samples were incubated with CoraLite®594-conjugated GM130 Polyclonal antibody (dilution 1:100, Proteintech, USA, Cat# CL594-11308) at 4 °C for 12 h to detect the Golgi. The cell nuclei DNA was stained with DAPI. The expression and colocalization of EGFR with FUT8 and Golgi were analyzed using a confocal laser scanning microscope (Leica Stellaris 5 WLL).

## Flow cytometry (FCM) analysis

For detecting the interaction between E2 and EGFR, Huh 7 cells were infected with HCV (MOI = 10) for the indicated time. And then cells were gently scraped off the bottom of the cell dish into the PBS using a cell scraper. For cell-surface staining, cell suspensions were washed twice in PBS and stained with the indicated antibodies for 30 min on ice and washed with PBS. For intracellular staining, the cells were fixed for 0.5 h (fixation buffer, BioLegend, Cat# 420801), permeabilized for 10 min (Perm Wash Buffer, BioLegend, Cat# 421002), and then stained with the indicated antibodies. All FCM analysis was conducted on CytoFlex (Beckman), and the data were analyzed using FlowJo V10, according to manufacturers' instructions. Antibodies used for flow cytometry analysis: HCV-E2 (dilution 1:50, SinoBiological, China, Cat# 40280-T62), EGFR (dilution 1:100, Proteintech, USA, Cat# 66455-1-Ig). Isotype control antibodies (Rabbit IgG control Polyclonal antibody, Proteintech, USA, Cat# 30000-0-AP; Mouse IgG1 isotype control monoclonal antibody, Proteintech, USA, Cat# 66360-1-Ig) were used to define background and non-specific binding signal.

## Detection of HCV RNA copies, HCV positive and negative strands RNA levels

HCV RNA copies were detected as follows: Absolute copies of HCV RNAs were determined using a diagnostic kit for a one-step HCV RT-qPCR (Sun Yat-Sen University Daan Gene Co. Ltd., China, Cat# DA-Z070) with the standard curve method using TaqMan probe according to the procedures provided by the manufacturer's instructions.

HCV positive (+) strands RNA levels were detected by RT-qPCR as follows: In brief, both HCV RNA and total cellular RNA were prepared from HCV-infected Huh7 cells or Huh7 cells/tissue lysates using TRIzol reagent (Invitrogen, USA, Cat# 15596026), according to the manufacturer's protocol. Briefly, the ReverTra Ace-First Strand cDNA Synthesis Kit was used to generate cDNAs from cellular mRNA and HCV RNA. Specific mRNAs and HCV RNA were quantified using the SYBR Green Real-Time PCR Master Mix (Toyobo, Japan, Cat# QPK-201) and the corresponding primers. The mRNA and HCV RNA levels were normalized to that of GAPDH. Relative RNA levels were calculated using the comparative cycle threshold (CT) method ($2^{-\Delta\Delta Ct}$, method), where CT represents the amplification cycle number at which the fluorescence generated within a reaction rises above a defined threshold, and $\Delta\Delta Ct$ = experimental groups ($Ct_{HCV} - Ct_{GAPDH}$) − control groups ($Ct_{HCV} - Ct_{GAPDH}$). The mRNA levels in the experimental groups are presented as the fold levels relative to the control groups, calculated with the following formula: $2^{-\Delta\Delta Ct}$. The sequences of the RT-qPCR primers are listed in Supplementary Table 3.

HCV negative (−) strand RNA levels were detected as follows: HCV RNA was extracted from HCV-infected Huh7 cells/tissue lysates using the TRIzol reagent (Invitrogen, USA, Cat# 15596026) according to the manufacturer's protocol. For Tth-based RT-qPCR detection of the HCV negative strand, cDNA was synthesized in 20 μL of reaction mixture containing 1 μM outer sense primer, 1 μg RNA, 20 mM MnCl$_2$, 2 mM (each) dNTP, and 5 U of Tth (Applied Biosystem, Life Technologies, Carlsbad, USA, Cat# TTH-301). After 20 min at 65 °C, Mn$^{2+}$ was chelated with 8 μL of 10-EGTA chelating buffer (100 mM Tris-HCL(pH=8.3),1 M KCL,7.5 mM EGTA,0.5% Tween-20), 1 μM outer anti-sense primer was added, the volume was adjusted to 100 μL, and the MgCl$_2$ concentration was adjusted to 2.2 mM. The general procedure for PCR was described as the following: 94 °C for 1 min, 20 cycles of (94 °C for 15 s, 60 °C for 30 s, and 72 °C for 15 s), 72 °C for 7 min. The quantification of HCV negative-strand RNAs was performed using SYBR Green real-time PCR Master Mix plus (QPK-212, Toyobo, Japan, Cat# QPK-212) for inner sense and inner anti-sense primers. The HCV RNA levels were normalized to those of GAPDH. Relative fold differences were determined using the method of delta Ct ($2^{-\Delta\Delta Ct}$ method). $\Delta\Delta Ct$ = FF-treated group ($Ct_{HCV} - Ct_{GAPDH}$) - Control group ($Ct_{HCV} - Ct_{GAPDH}$). All experiments were performed at least thrice. The primer sequences are as follows: HCV-F/outer sense: 5′- AATCACTCCCCTGTGAGGAAC and HCV-R/outer anti-sense: 5′- TGGTGCACGGTCTACGAGACCTC (R represents A or G; Y represents C or G)[98]; HCV inner sense: 5′-ACTGTCTTCACGCAGAAAGCGCC-3′; HCV anti-inner sense: 5′-CAAGCGCCCTATCAGGCAGTACC-3′[99].

## Statistical analysis

All data are presented as the mean ± standard deviation (SD) and analyzed by GraphPad Prism V.8.00 software (GraphPad Software, San Diego, CA, USA). Each group of data was subjected to the Shapiro-Wilk test for normal distribution. Student's $t$ test was used to determine the significance of differences between two groups of normally distributed data. For comparisons between multiple groups, ordinary one-way or two-way analysis of variance (ANOVA), followed by the Sidak's post hoc-test, was used. $p < 0.05$ was considered statistically significant.

## Reporting summary

Further information on research design is available in the Nature Portfolio Reporting Summary linked to this article.

## Data availability

The raw microarray data in this study were retrieved from Gene Expression Omnibus (accession code GSE42405). Source data are provided with this paper.

## Code availability

Code to perform analysis of microarray data is available on GitHub ([https://github.com/panqiu777/HE-TIPS-used-for-publication/releases/tag/v1.0], [https://doi.org/10.5281/zenodo.10431651]).

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

## Acknowledgements

This work was supported by grants from the National Natural Science Foundation of China (82230078, 22077097, 91740120, 21572173 and 21721005 to X.-L.Z.; 82272978 to M.L.), National Outstanding Youth Foundation of China (81025008 to X.-L.Z.), National Key R&D Program of China (2022YFA1303500, 2018YFA0507603 to X.-L.Z.), Medical Science Advancement Program (Basical Medical Sciences) of Wuhan University (TFJC 2018002 to X.-L.Z.), Key R&D Program of Hubei Province (2020BCB020 to X.-L.Z.), the Hubei Province's Outstanding Medical Academic Leader Program (523-276003 to X.-L.Z.), the Innovative Group Project of Hubei Health Committee (WJ2021C002 to X.-L. Z.), the Foundational Research Funds for the Central University of China (2042022dx0003, 2042023kf1011 to X.-L.Z.) and Natural Science Foundation Project of Hubei Province (2021CFB484 to M.L.).

## Author contributions

Q.P. conducted the experiments, analyzed data and interpreted results. Y.X. analyzed data. Y.Z., J.W. and X.-Q. Guo provided help to experiments. Q.P., Y.X., and M.L. wrote and revised the manuscript. X.-L. Zhang directed experimental design and the overall study, interpreted data, supervised research and revised the manuscript.

## Competing interests

The authors declare no competing interests.
