## [Peer Review File · Nature Communications]

Reviewers' Comments:

Reviewer #1:

Remarks to the Author:

This manuscript describes novel biological function of core-fucose/Fut8 in terms of viral infection. Infection of enveloped virus induced increases in Fut8 expression. The authors showed molecular mechanism by which FUT8 increases due to HCV infection, resulting in enhanced core fucosylation of EGFR and subsequently promoting the activation of STAT3 and degradation of RIG-1, ultimately leading to the suppression of IFN- β expression and increased HCV replication by a considerable amount of data. However, some of the data were not clear, and there were confusing explanations from the authors in some parts. The authors should clarify and amend the manuscript before publication as described below.

Major points

1. Since enveloped viruses can include both DNA and RNA viruses, so it is difficult to understand why similar effects are not observed in DNA viruses. HSV-1, which is a DNA virus, has been shown to induce FUT8 expression. Additionally, among enveloped viruses such as HCV, VSV, HSV-1, SARS-CoV-2, and HIV-gp120, the extent of FUT8 induction varies. However, it is not clear what differences in each envelope protein lead to these variations, nor is it indicated whether the receptors involved in infection are the same or different.
2. The authors consistently used AAL as a marker for core fucosylation induced by increased FUT8 expression, citing Ref.27. If they are citing this paper, they should use AOL at least. AOL and AAL are different lectins with characteristic recognition of fucosylation linkage.
3. In Fig.1, the authors investigated expression of fucosyltransferases (Futs) alone. In general, cellular fucosylation is regulated by GDP-fucose levels, expression of GDF-fucose transporter, and Futs. The authors should check expression of GMDS, FX, SLC35C1, and Futs in Fig.1 experiments.

Minor points;

1. L-178: In Figure 2I, it appears that FUT8 is upregulated by shSNAIL alone.
2. L-284: It should be "FUT8KO Huh7.5.1 cells complemented with exogenous RIG-1" instead of "WT Huh7.5.1 cells."
3. L-298: It should be "Figure 5E" and "Figure 5F" instead of "Figure 4E" and "Figure 4F," respectively.
4. L-308: In Figure 5H, "lane 8 vs 7" should be "lane 6 vs 5."
5. L-312: "VSV-induced" is unnecessary for "VSV-induced IFN-I."
6. L-323: It should be "C57BL/6J" instead of "FUT8KO ICR mice are embryonically lethal."
7. Figure 4A: It appears that His-RIG-I is reduced even without STAT3 activation, contrary to the authors' explanation.

Reviewer #2:

Remarks to the Author:

In this manuscript, Pan et al. characterize the role of N-glycan core fucosylation, mediated by alpha-(1,6)-fucosyltransferase (FUT8), in the replication of HCV and other RNA viruses. They first show that HCV, VSV, SARS-CoV-2 or HSV-1 infection, but not SeV or E. coli, induces expression of FUT8 with a corresponding increase in core fucosylation. The increase in FUT8 expression was attributed to EGFR activation by viral glycoproteins (HCV E2, VSV G, SARS-CoV-2 Spike) and subsequent signaling via AKT to activate the SNAIL transcription factor. Functionally, the increased FUT8 expression led to an increase in EGFR core fucosylation, which was required for downstream activation of the JAK1/STAT3 pathway. Finally, the authors identified a proviral role for FUT8-induced activation of EGFR/JAK1/STAT3, where activated STAT3 induces K48-linked ubiquitination and degradation of the pattern recognition receptor RIG-I, in a manner dependent on Trim40, thus inhibiting antiviral immune responses and promoting RNA virus replication. Consistently, in vivo experiments in mouse models of HCV and VSV infection showed that silencing of FUT8 inhibits viral replication by enhancing interferon responses.

Overall, the manuscript is comprehensive and presents compelling data supporting a proviral role for FUT8 in RNA virus infection via suppression of RIG-I-mediated antiviral responses. The data are technically sound and interpreted appropriately, for the most part (see comments below). These findings are novel and likely to be of interest for a broad audience, including virologists,

immunologists and cell biologists. Some considerations suggested below would further strengthen the manuscript:

Specific comments:

1. The authors state that Sendai virus (SeV) is non-enveloped (e.g., Line 94, 134, 384). However, SeV belongs to the Paramyxoviridae family and is in fact enveloped. The observation that SeV infection does not induce FUT8 expression may be attributed to lack of EGFR activation by SeV. Literature (e.g. Lupberger et al. (2011) Nat Med) shows that measles virus, a related Paramyxoviridae family member, does not require EGFR for infection. As a control, it would be interesting to test whether SeV activates EGFR.

2. Related to the above point, the role of EGFR activation in inducing FUT8 expression during viral infection could be more convincingly demonstrated for the viruses other than HCV. It would be good to show that VSV, SARS-CoV-2, HSV-1 (and SeV) or their corresponding glycoproteins activate (or not) EGFR, and that the increased FUT8 expression (e.g. as shown in Figure S1) can be blocked by silencing or inhibiting EGFR.

3. Figures 3C and 4E show robust RIG-I expression in HCV-infected cells, even with increased FUT8 expression levels (which should be induced by HCV, as shown in earlier figures). Are endogenous levels of FUT8 sufficient to induce RIG-I-degradation? Comparing RIG-I and FUT8 expression over a longer time course of HCV infection may be informative. Could other mechanisms that affect RIG-I activation (beyond its degradation) contribute to the phenotype?

Minor comments

1. What MOI was used for the cell culture viral infection experiments? It would be helpful to mention the MOI in the figure caption where applicable, and describe the experiments more thoroughly in the Methods section (particularly for VSV, SeV, HSV-1, where details are lacking).

2. Figure 3C: The labels on this blot are shifted to the left and not directly above the corresponding lanes

3. Line 298: Should be referring to figures 5E and 5F (not 4E/F).

4. Line 312: Likely should be "RIG-I-induced IFN-I" (not "VSV-induced IFN-I induced by other viruses")

5. Figure 6F: It's difficult to appreciate the increase in nuclear localization of pIRF3 from the immunohistochemical images. As such, would suggest removing statement about nuclear localization from the text (Line 340-341), unless nuclear localization were to be more explicitly tested.

6. Line 408-409: Figure 3D shows that glycosylation site-specific mutations caused by gene engineering do not affect protein synthesis, but what data supports the same claim made about "glycosylation inhibitors"?

7. Line 965: Figure 5 title "FUT8 promotes RNA viral throng" should be "FUT8 promotes RNA viral replication"?

8. Figure S1 caption includes panel (I) (Lines 1021-1023) but the corresponding panel is not shown in the figure.

9. Line 1027 – Supplementary Figure 2 caption title states that FUT-8 promotes HSV-1 replication but this is not supported by the data or text description

10. SARS-CoV-2 is referred to as "SARS-COV-2" throughout the manuscript. Would change to SARS-CoV-2 as this is the more widely accepted nomenclature.

Reviewer #3:

Remarks to the Author:

Qiu Pan et al. have described the regulation of FUT8 and interferon signaling mechanisms by envelope glycoproteins from selected viruses. The authors identified upregulated FUT8 by enveloped viral infection regulates RIG-I signaling mediated anti-RNA viral innate immune responses. FUT8 induces EGFR core fucosylation in activating EGFR-JAK1-Stat3 signaling. The results may explain the mechanism for immune escape of RNA viruses and FUT8 as a potential target for antiviral therapies. The manuscript is well written, and the data is clean. However, the authors need to carefully interpret results implicating the observations as specific for enveloped viruses. Overall, this work is thoughtfully carried out and well presented. The authors may please consider the following comments for clarity of the manuscript and rational for interpretation.

Specific comments

1. The results describe the use of enveloped viruses, but most of them focus upon the use of cell transfection. The HCV E2 full-length protein expressed in these cells would not likely be secreted or expressed on the cell surface. Therefore, the lipid envelope would unlikely be important. Further, the results do not adequately address the split role of the EGFR pathway (Jak-Stat/Akt).
2. The use of multiple viruses is somewhat distracting as their mechanisms of function may differ (i.e. secreted, cell surface vs. ER retained protein) and are not elaborated.
3. Some statements are not well supported by data (Fig. 2D).
4. P. 4, line 110: The authors stated that "Our recent study showed that HCV promotes FUT8 expression in Huh7.5.1 cells". They did not follow up further at this point in relation to the theme of the present study, excepting a short comment on p.11, line 309 regarding RIG-I deficiency. The authors should further elaborate this aspect and match or contrast with their present observations.
5. P. 5, line 113: What fold upregulation was observed as significant needs to mention in context?
6. P. 5, line 135-138: "We also investigated...cells" The authors may mention the comparative increase in FUT8 regulation by different viruses and the meaning for that difference.
7. P. 12, line 322: Established HCV infected mouse model is not very convincing and needs additional background support. How long does HCV mRNA replicate?
8. P. 12, lines 341-347: the sentences are difficult to follow for interpretation of the results. The authors should clarify how they conclude offered in the manuscript.
9. P. 14, line 382-386: It remains unclear how the lipid envelope is working here. Is the transfected HCV E2 sequence from ectopic expression of cells transported to cell surface, secreted, or retained in the ER of the cells due to its retention signal? If it is retained in cells in its native E2 form what role does the viral envelope play here?
10. The discussion section reads well and is easy to follow.

Minor issues

- a. The authors may please edit the manuscript for further clarity.
- b. P. 3, line 83: The authors may skip the word "common" mechanism utilized by viruses to enter cells.
- c. P. 9, line 39: the title may mention HCV genome replication to conform with the observations.
- d. Little organization or altering figure numbers and presentation may help for easy flow of the results to follow by the readers.
- e. P. 13, line 379-380: The line needs revision.
- f. The authors often used the words "suppress HCV replication. – it should sound better with viral RNA replication.

Reviewer #1

This manuscript describes novel biological function of core-fucose/Fut8 in terms of viral infection. Infection of enveloped virus induced increases in Fut8 expression. The authors showed molecular mechanism by which FUT8 increases due to HCV infection, resulting in enhanced core fucosylation of EGFR and subsequently promoting the activation of STAT3 and degradation of RIG-1, ultimately leading to the suppression of IFN- β expression and increased HCV replication by a considerable amount of data. However, some of the data were not clear, and there were confusing explanations from the authors in some parts. The authors should clarify and amend the manuscript before publication as described below.

Major points

1-1. Since enveloped viruses can include both DNA and RNA viruses, so it is difficult to understand why similar effects are not observed in DNA viruses. HSV-1, which is a DNA virus, has been shown to induce FUT8 expression.

Response: We thank the reviewer for the valuable comment. According to the editor's and the reviewer's suggestion, we have changed the words "enveloped viruses" to "hepatitis C virus" in the title.

Our results indicated that multiple enveloped viral envelope proteins (**Figure 1 and Figure S2**, including HCV-E2, VSV-G, SARS-CoV-2-spike, HIV-gp120) and enveloped HSV-1 induced upregulation of FUT8, and we further focused that HCV induced FUT8 upregulation through activation of EGFR-AKT-SNAIL. Further, we found that FUT8 induced activation of EGFR/JAK1/STAT3, where activated-STAT3 induced K48-linked ubiquitination and degradation of the pattern recognition receptor RIG-I, and suppressed downstream RIG-I-IRF3-IFN-I pathway, thus inhibited RNA viruses (**Figure 5**). However, we found that FUT8 overexpression had no effects on DNA virus replication (such as HSV-1, **Figure S5C**).

As RNA viruses activate the RIG-I-IRF3-IFN-I pathway, while DNA viruses activate the cGAS-STING-IRF3-IFN-I pathway, we have performed additional experiments assessing the effects of FUT8 on the cGAS-STING-IRF3-IFN-I pathway activation. The new results showed that p-STING and p-IRF3 expression was upregulated at 16 h post HSV-1 infection, but FUT8 overexpression did not affect cGAS, p-STING and p-IRF3 expression (new **Figure S5D**). HSV-1 infection induced upregulation of IFN- β , but FUT8 overexpression or knockdown did not affect HSV1-induced IFN- β expression (**Figure S5B**) and HSV-1 viral replication (**Figure S5C**). These new results showed that FUT8 overexpression did not affect cGAS-STING-IRF3-IFN-I pathway, thus did not affect DNA viral replication. We supplemented the relevant results (Lines 399-405) in the revised manuscript shown in red color.

Figure S5. (B and C) HeLa cells were transfected with the indicated plasmids for 48 h, and then infected with HSV-1 (MOI = 0.01) for 12 h. RT-qPCR analysis of *IFNβ* mRNA expression (B) and HSV-1 replication level (C). Data are normalized based on *GAPDH* for B and C. **(D)** HeLa cells were transfected with empty vector or pc3.1FUT8 for 48 h, and then infected with HSV-1 (MOI = 0.1) for the indicated time. Immunoblot analysis of FUT8, cGAS, STING, p-STING, IRF3 and p-IRF3 in HeLa cells.

1-2. Additionally, among enveloped viruses such as HCV, VSV, HSV-1, SARS-CoV2, and HIV-gp120, the extent of FUT8 induction varies. However, it is not clear what

differences in each envelope protein lead to these variations, nor is it indicated whether the receptors involved in infection are the same or different.

Response: Thanks for the kind suggestion. We agree with reviewer’s comment. Although these enveloped viruses, except SeV, induce FUT8, the extent of FUT8 induction varies. For example, at the same MOI (0.1), VSV induced about 25-fold (Figure 1H), while HCV induced about 12-fold FUT8 upregulation (Figure 1A). Different viral envelope proteins might engage different receptors, and they might coopt receptors with different degrees. We have added these descriptions in the revised manuscript shown in red color (Lines 492-496).

According to the reviewer’s suggestion, we have also performed additional experiments assessing the effects of different viral envelope proteins (HCV-E2, VSVG, SARS-CoV-2-spike, and HIV-gp120) on FUT8 expression in WT and EGFR KO Huh7 cells, and the new results showed that these viral envelope proteins all induced FUT8, core fucosylation (assessed by AAL lectin blot) and SNAIL upregulation in WT but not in EGFR KO cells (new Figure S3C-3D). Among these viral envelope proteins, VSV-G induced the highest level of FUT8, core fucosylation and SNAIL expression at 48 h post transfection (Figure S3C). We also found that these viral envelope proteins induced upregulation of p-EGFR and p-AKT in WT Huh7 cells at 12 h post transfection, but not in EGFR KO Huh7 cells (Figure S3D). These new results suggest that all these viral envelope proteins could induce EGFR-AKT-SNAIL-FUT8 pathway, but with different degrees of activation. We have added these new results in the revision shown in red color (Lines 215-225).

Figure S3. (C-D) Lectin blot for core fucosylation (C), and immunoblot analysis of SNAIL (C), FUT8 (C), p-EGFR (D), EGFR (D), p-AKT (D) and AKT (D) in WT or

EGFR KO Huh7 cells transfected with expression plasmids encoding the indicated viral envelope proteins. M (in C and D) indicates the molecular weight marker.

Comment 2.

The authors consistently used AAL as a marker for core fucosylation induced by increased FUT8 expression, citing Ref.27. If they are citing this paper, they should use AOL at least. AOL and AAL are different lectins with characteristic recognition of fucosylation linkage.

Response: Thanks for the valuable comment. According to the reviewer's suggestion, we have performed additional AOL lectin blot assay. Both AOL and AAL lectin blot assays showed that HCV infection induced core fucosylation (new **Figure 1B** in the revision). Albeit AOL shows higher affinity for alpha1,6-fucosylated oligosaccharides than AAL, a substantial body of literature indicates that besides AOL, AAL has also been widely used to detect changes in core fucosylation¹⁻⁷. Please see Lines 113-117 in the revision shown in red color.

Figure 1. (B) Lectins (AAL and AOL) and immunoblot analysis of core fucosylation and FUT8 in lysates of Huh7 cells infected with HCV (MOI = 0.1) for the indicated time.

Comment 3.

In Fig.1, the authors investigated expression of fucosyltransferases (Futs) alone. In general, cellular fucosylation is regulated by GDP-fucose levels, expression of

GDPfucose transporter, and Futs. The authors should check expression of GMDS, FX, SLC35C1, and Futs in Fig.1 experiments.

Response: Thank you for the helpful suggestion. According to the reviewer's suggestion, we have done new experiments assessing the cellular fucosylation regulated by the GDP-fucose levels, and the expression of GDP-fucose transporter after HCV/VSV infection. We examined the expression of GDP-mannose 4, 6-dehydratase (GMDS, catalyzing the first step in the synthesis of GDP-fucose from GDP-mannose), GDP-4-keto-6-deoxy-mannose-3, 5-epimerase-4-reductase (FX), solute Carrier Family 35 Member C1 (SLC35C1, a GDP-fucose transporter) after HCV/VSV infection by RT-qPCR. We found that no significant differences were observed for the expression of GMDS, FX, SLC35C1 in HCV-infected Huh7 cells or VSV-infected HEK293T cells (**Figure S2H-S2M**). These findings suggest that virus-induced core fucosylation upregulation is mainly induced by FUT8, but not regulated by GDP-fucose and GDPfucose transporter, and the upregulation of FUT8 plays a pivotal role in the increased core fucosylation after HCV and VSV infection. We have presented the new data in the revision shown in red color (**Figure S2H-S2M**, Lines 154-162).

Figure S2. (H-M) RT-qPCR analysis of *SLC35C1*, *FX*, *GMDS* mRNA expression in HCV-infected Huh7 cells (MOI = 0.1) (H-J) or VSV-infected HEK293T cells (MOI = 0.1) (K-M). Data are normalized based on *GAPDH* for H-M.

Minor points Comment

1.

L-178: In Figure 2I, it appears that FUT8 is upregulated by shSNAIL alone.

Response: Thanks for the comment. We have repeated the experiment, and the results confirmed that FUT8 is not upregulated by shSNAIL alone (**Figure 2I**). Please see the new results in the **Figure 2I**.

Figure 2. (I) Immunoblot analysis of FUT8, core fucosylation, and NS3 in lysates of Huh7 cells transfected with SNAIL shRNA (I) for 48 h, and then infected with HCV (MOI = 0.1) for the indicated time.

Comment 2.

L-284: It should be "FUT8KO Huh7.5.1 cells complemented with exogenous RIG-1" instead of "WT Huh7.5.1 cells."

Response: Thanks for the kind reminder. We sincerely apologize for our imprecise description. According to the reviewer's suggestions, we have changed "In FUT8 KO Huh7.5.1, cells complemented with exogenous FUT8, RIG-I increased HCV RNA replication in contrast to that in WT Huh7.5.1 cells" to "In FUT8 KO Huh7.5.1 cells complemented with exogenous FUT8, RIG-I increased HCV RNA replication in contrast to that in FUT8KO Huh7.5.1 cells complemented with exogenous RIG-I"

(**Figure 5B**, lane 12 vs.11).” in the revised manuscript. Please see Lines 355-356 in the revision shown in red color.

Comment 3.

L-298: It should be "Figure 5E" and "Figure 5F" instead of "Figure 4E" and "Figure 4F," respectively.

Response: Thanks for the kind reminder. We sincerely apologize for our imprecise description. According to the reviewer’s suggestions, we have changed “We observed similar results for VSV (**Figures 4E and S2F**) and SeV (**Figure 4F**)” to “We observed similar results for VSV (**Figures 5G and S5A**) and SeV (**Figure 5H**)” in the revised manuscript. Please see Lines 378 in the revision shown in red color.

Comment 4.

L-308: In Figure 5H, "lane 8 vs 7" should be "lane 6 vs 5."

Response: Thanks for the kind reminder. According to the reviewer’s suggestions, we have changed “**Figure 5H**, lane 8 vs. 7” to “**Figure 5F**, lane 6 vs. 5” in the revised manuscript. Please see Lines 368-369 in the revision shown in red color.

Comment 5.

L-312: "VSV-induced" is unnecessary for "VSV-induced IFN-I."

Response: Thanks for the kind reminder. We sincerely apologize for our typo error. According the reviewer’s suggestions, we have deleted the words “VSV-induced”, and changed “we determined the effects of FUT8 on VSV-induced IFN-I induced by other RNA viruses (VSV and SeV)” to “we determined the effects of FUT8 on IFN-I induced by other RNA viruses (VSV and SeV)” in the revised manuscript. Please see Lines 376 in the revision shown in red color.

Comment 6.

L-323: It should be "C57BL/6J" instead of "FUT8KO ICR mice are embryonically lethal."

Response: Thanks for the kind comment. Since we constructed ICR^{4R+} humanized mice for HCV infection based on ICR background mice, not with C57BL/6J mice. To make it clearer, we have changed “Fut8KO ICR mice are embryonically lethal and difficult to breed, we used *Fut8* tissue-specific knockdown or the FUT8 inhibitor 2FF in mouse infection models” to “Since Fut8KO in ICR background mouse is embryonically lethal and difficult to breed, we used *Fut8* tissue-specific knockdown in HCV-infected ICR^{4R+} transgenic mouse model or the FUT8 inhibitor 2FF in VSVinfected C57BL/6J mouse infection model”. We have added the above descriptions in the revision shown in red color (Lines 412-414).

Comment 7.

Figure 4A: It appears that His-RIG-I is reduced even without STAT3 activation, contrary to the authors' explanation.

Response: Thanks for the kind comment. In old **Figure 4A**, His-RIG-I is reduced even without STAT3 activation. The reason for the phenomenon is possibly due to the fact that ubiquitin overexpression induced mild RIG-I degradation in the absence of STAT3 overexpression (**Figure 4A**, lane 3 vs. 2). We have repeated the experiment, and the results confirmed that ubiquitin overexpression induced mild RIG-I degradation in the absence of STAT3 overexpression. We have added these descriptions in the revision shown in red color (Lines 281-282).

Figure 4. (A) STAT3 promotes K48 ubiquitination of RIG-I by Trim40. Huh7.5.1 cells were transfected with plasmids encoding His-RIG-I, HA-ubiquitin, Flag-STAT3, or siTrim40 (50 nM) for 24 h, and then infected with HCV (MOI = 0.1) for 48 h in the

presence or absence of MG132 (10 mM). Cells were harvested for immunoblot analysis of K48-Ub immunoprecipitated with antibody to His tag.

Reviewer #2

In this manuscript, Pan et al. characterize the role of N-glycan core fucosylation, mediated by alpha-(1,6)-fucosyltransferase (FUT8), in the replication of HCV and other RNA viruses. They first show that HCV, VSV, SARS-CoV-2 or HSV-1 infection, but not SeV or E. coli, induces expression of FUT8 with a corresponding increase in core fucosylation. The increase in FUT8 expression was attributed to EGFR activation by viral glycoproteins (HCV E2, VSV-G, SARS-CoV-2 Spike) and subsequent signaling via AKT to activate the SNAIL transcription factor. Functionally, the increased FUT8 expression led to an increase in EGFR core fucosylation, which was required for downstream activation of the JAK1/STAT3 pathway. Finally, the authors identified a proviral role for FUT8-induced activation of EGFR/JAK1/STAT3, where activated STAT3 induces K48-linked ubiquitination and degradation of the pattern recognition receptor RIG-I, in a manner dependent on Trim40, thus inhibiting antiviral immune responses and promoting RNA virus replication. Consistently, in vivo experiments in mouse models of HCV and VSV infection showed that silencing of FUT8 inhibits viral replication by enhancing interferon responses.

Overall, the manuscript is comprehensive and presents compelling data supporting a proviral role for FUT8 in RNA virus infection via suppression of RIG-I-mediated antiviral responses. The data are technically sound and interpreted appropriately, for the most part (see comments below). These findings are novel and likely to be of interest for a broad audience, including virologists, immunologists and cell biologists.

Some considerations suggested below would further strengthen the manuscript:

Specific comments:

Comment 1.

The authors state that Sendai virus (SeV) is non-enveloped (e.g., Line 94, 134, 384). However, SeV belongs to the Paramyxoviridae family and is in fact enveloped. The observation that SeV infection does not induce FUT8 expression may be attributed to lack of EGFR activation by SeV. Literature (e.g. Lupberger et al. (2011) Nat Med) shows that measles virus, a related Paramyxoviridae family member, does not require EGFR for infection. As a control, it would be interesting to test whether SeV activates EGFR.

Response: We thank the reviewer for the valuable comment. SeV is indeed an enveloped virus. According to the reviewer’s suggestion, we have performed new experiments assessing whether SeV activates EGFR, and we found that SeV infection had no effect on core fucosylation, FUT8 or p-EGFR-SNAIL activation (new **Figure S3E**). This result suggests that SeV, albeit an enveloped virus, does not induce FUT8 expression, which may be attributable to the lack of EGFR activation by SeV infection. We have changed the title and descriptions as the reviewer suggested. We have presented the new data (Lines 227-229) in the revision shown in red color

Figure S3. (E) Lectin blot analysis of core fucosylation and immunoblot analysis of FUT8, p-EGFR, EGFR and SNAIL in SeV-infected HEK293 cells (MOI = 0.1) for the indicated time.

Comment 2.

Related to the above point, the role of EGFR activation in inducing FUT8 expression during viral infection could be more convincingly demonstrated for the viruses other than HCV. It would be good to show that VSV, SARS-CoV-2, HSV-1 (and SeV) or their

corresponding glycoproteins activate (or not) EGFR, and that the increased FUT8 expression (e.g. as shown in Figure S1) can be blocked by silencing or inhibiting EGFR.

Response: We thank the reviewer for the valuable comment. According to the reviewer's suggestion, we have performed additional experiments assessing the effects of different viral envelope proteins (HCV-E2, VSV-G, SARS-CoV-2-spike, and HIVgp120) on FUT8 expression in WT and EGFR KO Huh7 cells, and the results showed that these viral envelope proteins all induced FUT8, core fucosylation (assessed by AAL lectin blot) and SNAIL upregulation in WT but not in EGFR KO cells (**Figure S3C-3D**). Among these viral envelope proteins, VSV-G induced the highest level of FUT8, core fucosylation and SNAIL expression at 48 h post transfection (**Figure S3C**). We also found that these viral envelope proteins induced upregulation of p-EGFR and p-AKT in WT at 12 h post transfection, but not in EGFR KO cells (**Figure S3D**). We have added these new results in the revision shown in red color (Lines 215-225).

Figure S3. (C-D) Lectin blot for core fucosylation (C), and immunoblot analysis of SNAIL (C), FUT8 (C), p-EGFR (D), EGFR (D), p-AKT (D) and AKT (D) in WT or EGFR KO Huh7 cells transfected with plasmids encoding the indicated viral envelope proteins. M (in C and D) indicates the molecular weight marker.

Comment 3.

Figures 3C and 4E show robust RIG-I expression in HCV-infected cells, even with increased FUT8 expression levels (which should be induced by HCV, as shown in earlier figures). Are endogenous levels of FUT8 sufficient to induce RIG-I-degradation? Comparing RIG-I and FUT8 expression over a longer time course of HCV infection may be informative.

Response: We thank the reviewer for the valuable comment. According to the reviewer's suggestion, we have done new experiments comparing RIG-I and FUT8 expression over a longer time course of HCV infection. As shown in new Figure 3E, HCV infection led to RIG-I increase at early stage (8 h.p.i.) and FUT8/core fucosylation upregulation at late stage (48-72 h.p.i.). Subsequently, a decrease in RIG-I protein level was observed after 48-72 h.p.i., since increased-FUT8 induced RIG-I degradation at late stage (new **Figure 4F**). But the RIG-I protein level at 72 h.p.i. was still higher than that at 0 to 4 h.p.i., due to HCV infection. This time course experimental result suggests that endogenous RIG-I could be induced by HCV infection at early stage (8 h.p.i.), and then partly degraded due to the HCV-induced FUT8 expression at late stage (48-72 h.p.i.). RIG-I was induced by virus infection, and endogenous FUT8 and RIG-I in Huh7 cells were lowly expressed in the absence of HCV infection (**Figure 4E**, lane 1; new **Figure 4F**, lane 1), so the FUT8-induced RIG-I degradation effects could be ignored in the absence of HCV infection. We have presented these new data in the Results (Lines 303-319) in the revised manuscript shown in red color.

F

Figure 4. (F) Lectin and immunoblot analysis of core fucosylation, FUT8 and RIG-I in lysates of Huh7 cells infected with HCV (MOI = 0.1) for the indicated time.

Could other mechanisms that affect RIG-I activation (beyond its degradation) contribute to the phenotype?

Response: We thank the reviewer for the valuable comment. According to the reviewer's suggestion, we have added some discussion about the other potential

mechanisms that affect RIG-I activation (beyond its degradation). Other studies have shown that deubiquinases (such as USP21), LGP2, and PKC α/β could suppress RIG-I activation⁸. Whether FUT8 regulates these pathways to induce RIG-I degradation needs further investigation. We have added the above descriptions in the Discussion in the revision shown in red color (Lines 538-541).

Minor comments: Comment

1.

What MOI was used for the cell culture viral infection experiments? It would be helpful to mention the MOI in the figure caption where applicable, and describe the experiments more thoroughly in the Methods section (particularly for VSV, SeV, HSV1, where details are lacking).

Response: We thank the reviewer for the valuable comment. According to the reviewer's suggestion, we have added the MOI value for the cell culture viral infection experiments in **Figure Legends** and **Methods** in the revised manuscript shown in red color. We have also added the following description about the infection experiments about VSV, SeV, HSV-1 in the **Methods** section of the revision shown in red color "For assessing the effects of viruses or bacteria on FUT8 expression, in brief, HEK293/Hela cells in 24-well plates were infected with VSV/SeV/HSV-1 (MOI = 0.1) or SARS-CoV-2 pseudovirus (MOI = 2) at 37 °C for 4 h or *E. coli* DH5 α (bacterium:cell = 100:1) at 37 °C for 1 h. The supernatants were discarded, and the infected cells were washed twice with PBS and incubated in DMEM containing 10% FBS for each experiment." (Lines 589-593).

Comment 2.

Figure 3C: The labels on this blot are shifted to the left and not directly above the corresponding lanes.

Response: Thanks for the kind reminder. We sincerely apologize for our carelessness. We have corrected this error and checked all the figures for the positioning of the labels.

Comment 3.

Line 298: Should be referring to figures 5E and 5F (not 4E/F).

Response: Thanks for the kind reminder. We sincerely apologize for our carelessness. According to the reviewer's suggestions, we have changed "We observed similar results for VSV (Figures 4E and S2F) and SeV (Figure 4F)" to "We observed similar results for VSV (Figures 5G and S5A) and SeV (Figure 5H)" in the revised manuscript. Please see Line 378 in the revision shown in red color.

Comment 4.

Line 312: Likely should be "RIG-I-induced IFN-I "(not "VSV-induced IFN-I induced by other viruses")

Response: Thanks for the kind reminder. We sincerely apologize for our carelessness. According to the Reviewer #1's and Reviewer #2's suggestions, we have deleted the words "VSV-induced", and changed "we determined the effects of FUT8 on VSV-induced IFN-I induced by other RNA viruses (VSV and SeV)" to "we determined the effects of FUT8 on IFN-I by other RNA viruses (VSV and SeV)" in the revised manuscript. Please see Lines 376 in the revision shown in red color.

Comment 5.

Figure 6F: It's difficult to appreciate the increase in nuclear localization of pIRF3 from the immunohistochemical images. As such, would suggest removing statement about nuclear localization from the text (Line 340-341), unless nuclear localization were to be more explicitly tested.

Response: Thanks for the valuable comment. According to the reviewer's suggestions, we have removed the statement about nuclear localization from the text. Please see Lines 434 in the revision shown in red color.

Comment 6.

Line 408-409: Figure 3D shows that glycosylation site-specific mutations caused by gene engineering do not affect protein synthesis, but what data supports the same claim made about “glycosylation inhibitors”?

Response: Thanks for the kind reminder. We sincerely apologize for our misdescription. In the revised manuscript, we remove the statement about “glycosylation inhibitors” (Line 520).

Comment 7.

Line 965: Figure 5 title “FUT8 promotes RNA viral throng” should be “FUT8 promotes RNA viral replication”?

Response: Thanks for the kind comment. We have changed “FUT8 promotes RNA viral throng and suppresses RIG-I-induced type I IFN production” to “FUT8 promotes RNA viral replication and suppresses RIG-I-induced type I IFN production” in the revised manuscript. Please see Line 1188 (Figure 5 title) in the revision shown in red color.

Comment 8.

Figure S1 caption includes panel (I) (Lines 1021-1023) but the corresponding panel is not shown in the figure.

Response: Thanks for the kind reminder. We sincerely apologize for our carelessness. In the revised manuscript, we have deleted the words “(I) Huh7 cells were infected with HCV for 48h. Cells were fixed and labelled for EGFR, FUT8 and the Golgi marker GM130. DAPI was used to stain nuclei. Representative confocal microscopy images are shown.” in original Figure S1 caption (the new Figure S2 caption).

Comment 9.

Line 1027 – Supplementary Figure 2 caption title states that FUT-8 promotes HSV-1 replication but this is not supported by the data or text description

Response: Thanks for the kind reminder. We sincerely apologize for our typo error. It should be VSV. In the revised manuscript, we have substituted “VSV” for “HSV-1”.

We have changed the Supplementary Figure 2 caption title as “**Related to Figure 2.**” in the revised manuscript shown in red color (Line 1271).

Comment 10.

SARS-CoV-2 is referred to as “SARS-COV-2” throughout the manuscript. Would change to SARS-CoV-2 as this is the more widely accepted nomenclature.

Response: Thanks for the valuable comment. According to the reviewer’s suggestion, we have changed “SARS-COV-2” to “SARS-CoV-2” in the revised manuscript shown in red color throughout the manuscript.

Reviewer #3

Qiu Pan et al. have described the regulation of FUT8 and interferon signaling mechanisms by envelope glycoproteins from selected viruses. The authors identified upregulated FUT8 by enveloped viral infection regulates RIG-I signaling mediated anti-RNA viral innate immune responses. FUT8 induces EGFR core fucosylation in activating EGFR-JAK1-Stat3 signaling. The results may explain the mechanism for immune escape of RNA viruses and FUT8 as a potential target for antiviral therapies. The manuscript is well written, and the data is clean. However, the authors need to carefully interpret results implicating the observations as specific for enveloped viruses. Overall, this work is thoughtfully carried out and well presented. The authors may please consider the following comments for clarity of the manuscript and rational for interpretation.

Specific comments:

Comment 1.

The results describe the use of enveloped viruses, but most of them focus upon the use of cell transfection. The HCV E2 full-length protein expressed in these cells would not likely be secreted or expressed on the cell surface. Therefore, the lipid envelope would unlikely be important.

Response: Thank you for the helpful suggestion. Previous reports have shown that fulllength HCV-E2 contains transmembrane region (references shown below)^{9,10}, usually could be expressed on the cell surface. We have also performed new experiments, and observed similar results. Confocal microscopy analysis showed that HCV-E2 (red) colocalized with cellular endogenous EGFR (green) on the cell surface of Huh7 cells transfected with pcDNA3.1-myc-His-E2 (co-localization indicated by the orange color in **Figure S3B**), but not in pcDNA3.1 empty vector group. We have presented the new data in the revised manuscript shown in red color (**Figure S3B**; Lines 209-214).

Figure S3. (B) Huh7 cells were transfected with HCV E2 plasmid or empty vector for 48 h. Cells were fixed and probed for EGFR and His-E2. DAPI was used to stain cellular nuclei. Representative confocal microscopy images are shown.

Further, the results do not adequately address the split role of the EGFR pathway (JakStat/Akt).

Response: Thank you for the helpful suggestion. According to the reviewer's suggestion, we have performed additional experiments to reveal the relationship between HCV E2-EGFR-p-AKT-SNAIL-FUT8 pathway and fucosylated-EGFR-pJAK-p-STAT3 pathway. We found that HCV infection induced FUT8-p-EGFR-pJAK1-p-STAT3 activation at 24 h, but this promoting effect disappeared after

knockdown of AKT (as shown below in **Figure S4E**). Our study reveals that HCV engaged EGFR to activate p-AKT-SNAIL pathway at early stage (**Figure 2D**, p-AKT peaked at 2 h.p.i. and remarkably decreased at 48 h.p.i.), thus induced FUT8 expression (**Figures 3E and S4E**, at 24-72 h.p.i.). Subsequently FUT8 induced fucosylatedEGFR-p-JAK-p-STAT3 activation after 24 h.p.i. (**Figures 3C and S4E**). We also demonstrated that FUT8 inhibitor 2FF could suppress the fucosylated-EGFR-p-JAKp-STAT3 activation (**Figure 3C**), suggesting that increased-FUT8 is the initiator of fucosylated-EGFR-p-JAK-p-STAT3 pathway. So, our results strongly suggest that HCV E2-p-EGFR-p-AKT-SNAIL-FUT8 pathway is at the upstream of fucosylatedEGFR-p-JAK-p-STAT3 pathway. We have added these descriptions in the Result section in the revision shown in red color (Lines 310-320).

Figure S4 (E) Huh7 cells were transfected with shAKT or shScramble for 48 h, and then infected with HCV for 0-72 h (MOI = 0.1). Lectin blot for core fucosylation and immunoblot analysis of p-EGFR, p-JAK1, p-STAT3, FUT8 and AKT in Huh7 cells.

Comment 2.

The use of multiple viruses is somewhat distracting as their mechanisms of function may differ (i.e. secreted, cell surface vs. ER retained protein) and are not elaborated.

Response: Thank you for the helpful suggestion. We agree with the reviewer that multiple viruses might differ in their mechanisms of action (i.e. the virus-EGFR interaction). Our article focused on HCV in the whole work, albeit we also found that

other enveloped viruses besides HCV (VSV, SARS-CoV-2, HIV, HSV-1) induced FUT8 upregulation. And we found that FUT8-induced RIG-I degradation and IFN-I suppression might be universal and critical for RNA viral replication suppression (such as HCV, VSV and SeV). According to the editor and reviewer's suggestion, we have modified the words "enveloped viruses" as "hepatitis C virus" in the title ("**EGFR core fucosylation, induced by hepatitis C virus, promotes TRIM40-mediated-RIG-I ubiquitination and suppresses interferon-I antiviral defenses**") and several descriptions in the revision shown in red color (Lines 215-225).

Comment 3.

Some statements are not well supported by data (Fig. 2D).

Response: Thank you for the helpful suggestion. According to the reviewer's suggestion, we have modified the description of the Fig. 2D results, and have changed the original statement "HCV infection caused sequential activation of EGFR and AKT and an increase in SNAIL expression (Figure 2D)" to "We found that p-EGFR was activated at 1 h post HCV infection, reaching its peak at 2 h (**Figure 2D**). The p-AKT was initially activated at 2 h, and began to decrease at 6 h. The transcription factor SNAIL began to increase at 6 h. Subsequently, an increase in FUT8 expression was observed at 48 h (**Figure 2D**). This data indicate that HCV infection causes HCV E2-EGFR-p-AKT-SNAIL-FUT8 axis sequential activation (**Figure 2D**)."

We further determined the relationship between HCV E2-EGFR-p-AKT-SNAIL-FUT8 pathway and fucosylated-EGFR-p-JAK-p-STAT3 pathway, and found that HCV engaged EGFR to activate p-AKT-SNAIL pathway at early stage (**Figure 2D**, p-AKT peaked at 2 h.p.i. and remarkably decreased at 48 h.p.i.), and thus induced FUT8 expression (**Figure 3E**, after 12-72 h.p.i.), and subsequently increased FUT8-induced fucosylated-EGFR-p-JAK-p-STAT3 activation occurred (**Figure 3C**). We also demonstrated that FUT8 inhibitor 2FF could suppress the fucosylated-EGFR-p-JAK-p-STAT3 activation (**Figure 3C**), suggesting that increased-FUT8 is the initiator of fucosylated-EGFR-p-JAK-p-STAT3 pathway. So, our results strongly suggest that HCV E2-EGFR-p-AKT-SNAIL-FUT8 pathway is at the upstream of fucosylated-EGFR-p-

JAK-p-STAT3 pathway. We have added the above descriptions in the revised manuscript shown in red color (Lines 195-200, 310-320).

Comment 4.

P. 4, line 110: The authors stated that “Our recent study showed that HCV promotes FUT8 expression in Huh7.5.1 cells”. They did not follow up further at this point in relation to the theme of the present study, excepting a short comment on p.11, line 309 regarding RIG-I deficiency. The authors should further elaborate this aspect and match or contrast with their present observations.

Response: Thank you for the helpful suggestion. According to the reviewer’s suggestion, we have modified our description in the revision. Regarding “P. 4, line 110:”, we have added the following statements: “we further examined and confirmed that HCV promoted FUT8 expression in Huh7 cells, besides Huh7.5.1 cells (**Figure 1A**)”.

Regarding “p.11, line 309”, we have added the following statement “The results in **Figures 5B and 5D** showed that FUT8 protein level has no effect on IFN- β expression (**Figure 5D**) and HCV RNA replication (**Figure 5B**) in Huh7.5.1. However, FUT8 inhibited IFN- β expression and promoted HCV RNA replication in Huh7.5.1 cells complemented with exogenous RIG-I (**Figures 5B and 5D**). Results from RIG-I-rescued Huh7.5.1 cells (**Figures 5B, 5C and 5D**) were consistent with those in Huh7 cells (**Figures 5A, 5E and 5F**). ” Please see the Result section in the revision shown in red color (Lines 109-110, 360-363, 369-370).

Comment 5.

P. 5, line 113: What fold upregulation was observed as significant needs to mention in context?

Response: Thank you for the helpful suggestion. According to the reviewer’s suggestion, we have added the description of fold upregulation in the revision: “Among these genes, the mRNA level of *FUT8* was significantly upregulated 3- and 10-fold,

respectively, at 6 and 12 h post-infection of HCV (**Figure 1A**).” Please see the Result section in the revision shown in red color (Lines 112-113).

Comment 6.

P. 5, line 135-138: “We also investigated...cells” The authors may mention the comparative increase in FUT8 regulation by different viruses and the meaning for that difference.

Response: We appreciate the reviewer’s suggestion. According to the reviewer’s suggestion, we have performed additional experiments assessing the effects of different viral envelope proteins (HCV-E2, VSV-G, SARS-CoV-2-spike, and HIV-gp120) on FUT8 expression in WT and EGFR KO Huh7 cells, and the results showed that these viral envelope proteins all induced FUT8, core fucosylation (assessed by AAL lectin blot) and SNAIL upregulation in WT but not in EGFR KO cells (**Figure S3C-3D**). Among these viral envelope proteins, VSV-G induced the highest level of FUT8, core fucosylation and SNAIL expression at 48 h post transfection (**Figure S3C**), followed by SARS-CoV-2-Spike, HCV-E2 and HIV-gp120. We also found that these viral envelope proteins induced upregulation of p-EGFR and p-AKT in WT at 12 h post transfection, but not in EGFR KO cells (**Figure S3D**). We have added these new results in the revision shown in red color (Lines 215-225).

Figure S3. (C-D) Lectin blot for core fucosylation (C), and immunoblot analysis of SNAIL (C), FUT8 (C), p-EGFR (D), EGFR (D), p-AKT (D) and AKT (D) in WT or EGFR KO Huh7 cells transfected with plasmids encoding the indicated viral envelope proteins. M (in C and D) indicates the molecular weight marker.

Comment 7.

P. 12, line 322: Established HCV infected mouse model is not very convincing and needs additional background support. How long does HCV mRNA replicate?

Response: Thanks for the valuable comment. We have supplemented HCV infection mouse model background supporting data. According to the reviewer's suggestion, we have performed new experiments, and have added the following new data and new descriptions in the revision "We utilized a humanized HCV infection mouse model, which harbored human scavenger receptor B1 (SR-B1), CD81, claudin-1 (CLDN1), and occluding (OCLN) (essential receptors or coreceptors for HCV cell entry) genes (**Figure S1A**), named as ICR^{4R+} mice. As shown in **Figure S1B**, both ICR^{4R+} and ICR background parental mice were infected with HCV. We detected that serum HCV particle copies, liver HCV RNA positive (+) and negative (-) strand replication continuously increased and peaked at Day 42, and then maintained at high levels at least during our detection period (for 56 days) in ICR^{4R+} mice but not in HCV-infected parental ICR mice (**Figure S1C-E**). The liver function test alanine transaminase (ALT) levels (indicating the level of liver damage) also significantly increased at Day 49 post infection in ICR^{4R+} mice but not in HCV-infected parental ICR mice (**Figure S1F**). Immunohistochemistry results also showed that HCV Core protein expression was observed at Day 49 post infection in livers from ICR^{4R+} mice but not HCV-infected parental ICR mice (**Figure S1G**). Similar results were reported in previous report using the HCV-infected ICR^{2R+} (transgenic mice in ICR background harboring both human CD81 and occludin genes) mouse model¹¹." Please see the new **Figure S1** and **Results** section in the revision shown in red color (Lines 118-131).

Figure S1

Figure S1. ICR^{4R+} transgenic mouse model for HCV infection. (A) Immunoblot analysis of human SR-BI, OCLN, CD81, and CLDN-1 in the liver tissues of ICR^{4R+}/ICR background mice. (B) Scheme of ICR^{4R+} transgenic mouse model for HCV infection (n = 3 for each time point in each group, 48 mice in total). (C) The blood samples were collected at different time points during the course of HCV infection. Mouse serum HCV RNA absolute copies were determined using RT-qPCR with standard curve method using TaqMan probe. (D-E) Analysis of relative HCV-negative strand (-) RNA levels using strand-specific Tth-based RT-qPCR (D), and relative

HCVpositive strand (+) RNA levels by RT-qPCR analysis (E) in the liver tissues of ICR/ICR^{4R+} mice infected with HCV. (F) Serum ALT levels were measured in ICR/ICR^{4R+} mice at Day 49 post infection. All experiments were performed in triplicate. (G) H&E stain (upper panel) and immunohistochemical staining using anti-Core (lower panel, brown color indicates positively stained region) in liver tissues of HCV-infected ICR/ICR^{4R+} mice. Data are normalized based on *Gapdh* for C-E. Data in all quantitative panels are presented as mean \pm SD. Two-tailed unpaired student's t test was used to assess the statistical difference in C-F. **** $p < 0.0001$. ns: no significant difference.

Comment 8.

P. 12, lines 341-347: the sentences are difficult to follow for interpretation of the results. The authors should clarify how they conclude offered in the manuscript.

Response: We thank the reviewer for this comment. According to the reviewer's suggestion, we have added the following descriptions in the Result section:

“Liver inflammatory infiltration is a histological feature, usually representing that immune cells have been recruited to the liver during HCV infection^{12 13}. The presence and nature of portal inflammatory infiltrates can help assess the extent of liver damage and inflammation caused by the virus. The boundary between the white pulp and red pulp in the spleen can become blurred, and lymphoid tissue in the white pulp may undergo hyperplasia (an increase in cell numbers) appearing as large masses within the spleen, due to the inflammation and damage.^{14,15} Histopathological examination of the tissues (**Figures S6B and S6C**) revealed the following: (a) Portal inflammatory infiltrates (inside dashed white circles) were observed in PBS plus HCV group and shScramble plus HCV group, but fewer in sh-Fut8 plus HCV group and none in PBS group (**Figure S6B**). (b) As a result of HCV infection, the white pulps joined to form a large mass with blurred boundary between white pulp and red pulp in PBS plus HCV group and shScramble plus HCV group, but these changes were less pronounced in sh-Fut8 plus HCV group and none in only PBS group (**Figure S6C**). These data strongly suggest that hepatic-targeted *Fut8* silencing suppresses HCV RNA replication, and alleviates inflammation and tissue damage *in vivo*.”

We have added the above descriptions in the revision shown in red color (Lines 435448).

Comment 9.

P. 14, line 382-386: It remains unclear how the lipid envelope is working here. Is the transfected HCV E2 sequence from ectopic expression of cells transported to cell surface, secreted, or retained in the ER of the cells due to its retention signal? If it is retained in cells in its native E2 form what role does the viral envelope play here?

Response: Thank you for the helpful suggestion. Previous reports have shown that fulllength HCV-E2 contains transmembrane region (references shown below)^{9,10}, usually could be expressed on the cell surface. We have also performed new experiments, and observed similar results. Confocal microscopy analysis showed that HCV-E2 (red) colocalized with cellular endogenous EGFR (green) on the cell surface of Huh7 cells transfected with pcDNA3.1-myc-His-E2 (co-localization indicated by the orange color in **Figure S3B**), but not in pcDNA3.1 empty vector group. We have presented the new data in the revised manuscript shown in red color (**Figure S3B**; Lines 209-214).

Figure S3. (B) Huh7 cells were transfected with HCV E2 plasmid or empty vector for 48 h. Cells were fixed and probed for EGFR and His-E2. DAPI was used to stain cellular nuclei. Representative confocal microscopy images are shown.

Comment 10.

The discussion section reads well and is easy to follow.

Response: Thanks for the kind comment. We appreciate your comment.

Minor issues:

Comment a.

The authors may please edit the manuscript for further clarity.

Response: Thanks for the kind comment. According to reviewer's suggestion, we have edited the manuscript for clarity (shown in red color in the revision).

Comment b.

P. 3, line 83: The authors may skip the word "common" mechanism utilized by viruses to enter cells.

Response: Thank you for the helpful suggestion. In the revised manuscript, we have deleted the "common" (Line 83).

Comment c.

P. 9, line 239: the title may mention HCV genome replication to conform with the observations.

Response: Thank you for the helpful suggestion. In the revised manuscript, we have changed from "HCV replication" to "HCV RNA replication". Please see P11, Line 298 in the revision shown in red color.

Comment d.

Little organization or altering figure numbers and presentation may help for easy flow of the results to follow by the readers.

Response: Thank you for the helpful suggestion. According to your suggestion, we have modified the organization, and altered figure numbers and presentation to enhance

clarity. For example, the original **Figure S2** was divided into new **Figure S4** and **Figure S5** to facilitate easy flow of the results.

Comment e.

P. 13, line 379-380: The line needs revision.

Response: Thank you for the helpful suggestion. According to your suggestion, we have modified the original sentence from “More attention has been paid to the changes in viral life cycles caused by viral glycosylation modification of viruses. However only a few reports have focused on the modification of the host cell glycosylation profile and activation of host cell glycosyltransferase transcription during viral infection” to “Up till now, only a few reports have focused on the modification of the host cell glycosylation profile and activation of host cell glycosyltransferase transcription during viral infection, although extensive studies have investigated the impact of viral glycosylation modification on viral life cycles”. Please see Lines 482-485 in the revision shown in red color.

Comment f.

The authors often used the words “suppress HCV replication. – it should sound better with viral RNA replication.

Response: Thank you for the helpful suggestion. In the revised manuscript, we have changed from “HCV replication” to “HCV RNA replication” throughout the manuscript in the revision shown in red color.

References for Reply

- 1 Matsumura, K. *et al.* Carbohydrate binding specificity of a fucose-specific lectin from *Aspergillus oryzae*: a novel probe for core fucose. *J Biol Chem* **282**, 15700-15708 (2007). <https://doi.org:10.1074/jbc.M701195200>
- 2 Iijima, J. *et al.* Core fucose is critical for CD14-dependent Toll-like receptor 4 signaling. *Glycobiology* **27**, 1006-1015 (2017). <https://doi.org:10.1093/glycob/cwx075>
- 3 Wang, X. *et al.* Dysregulation of TGF-beta1 receptor activation leads to abnormal lung development and emphysema-like phenotype in core

- fucosedeficient mice. *Proc Natl Acad Sci U S A* **102**, 15791-15796 (2005). <https://doi.org:10.1073/pnas.0507375102>
- 4 Lin, S. *et al.* Alpha-(1,6)-fucosyltransferase (FUT8) affects the survival strategy of osteosarcoma by remodeling TNF/NF- κ B2 signaling. *Cell Death Dis* **12**, 1124 (2021). <https://doi.org:10.1038/s41419-021-04416-x>
- 5 Pieri, V. *et al.* Aberrant L-Fucose Accumulation and Increased Core Fucosylation Are Metabolic Liabilities in Mesenchymal Glioblastoma. *Cancer Res* **83**, 195-218 (2023). <https://doi.org:10.1158/0008-5472.CAN-22-0677>
- 6 Hayashiji, N. *et al.* α -1,6-Fucosyltransferase Is Essential for Myogenesis in Zebrafish. *Cells* **12** (2022). <https://doi.org:10.3390/cells12010144>
- 7 Kurimoto, A. *et al.* The absence of core fucose up-regulates GnT-III and Wnt target genes: a possible mechanism for an adaptive response in terms of glycan function. *J Biol Chem* **289**, 11704-11714 (2014). <https://doi.org:10.1074/jbc.M113.502542>
- 8 Reikine, S., Nguyen, J. B. & Modis, Y. Pattern Recognition and Signaling Mechanisms of RIG-I and MDA5. *Front Immunol* **5**, 342 (2014). <https://doi.org:10.3389/fimmu.2014.00342>
- 9 Krey, T. *et al.* The disulfide bonds in glycoprotein E2 of hepatitis C virus reveal the tertiary organization of the molecule. *PLoS Pathog* **6**, e1000762 (2010). <https://doi.org:10.1371/journal.ppat.1000762>
- 10 Koutsoudakis, G. *et al.* Interplay between basic residues of hepatitis C virus glycoprotein E2 with viral receptors, neutralizing antibodies and lipoproteins. *PLoS One* **7**, e52651 (2012). <https://doi.org:10.1371/journal.pone.0052651>
- 11 Chen, J. *et al.* Persistent hepatitis C virus infections and hepatopathological manifestations in immune-competent humanized mice. *Cell Res* **24**, 1050-1066 (2014). <https://doi.org:10.1038/cr.2014.116>
- 12 Chen, S. R. *et al.* Celastrol attenuates hepatitis C virus translation and inflammatory response in mice by suppressing heat shock protein 90 β . *Acta Pharmacol Sin* **44**, 1637-1648 (2023). <https://doi.org:10.1038/s41401-02301067-w>
- 13 Weber, A., Boege, Y., Reisinger, F. & Heikenwalder, M. Chronic liver inflammation and hepatocellular carcinoma: persistence matters. *Swiss Med Wkly* **141**, w13197 (2011). <https://doi.org:10.4414/smw.2011.13197>
- 14 Lewis, S. M., Williams, A. & Eisenbarth, S. C. Structure and function of the immune system in the spleen. *Sci Immunol* **4** (2019). <https://doi.org:10.1126/sciimmunol.aau6085>
- 15 Bhatia, K., Sahdev, A. & Reznick, R. H. Lymphoma of the spleen. *Semin Ultrasound CT MR* **28**, 12-20 (2007). <https://doi.org:10.1053/j.sult.2006.10.010>

Reviewers' Comments:

Reviewer #1:

Remarks to the Author:

The authors revised their manuscript very well. Just one point should be amended.

In the authors' response to comment 6, there is a statement that "Fut8KO ICR mice are embryonically lethal and difficult to breed" but the embryonically lethal mice are C57BL/6J mice, not ICR mice. Fujii H et al. conducted research using ICR Fut8KO mice. Please refer to Gastroenterology 2016. Therefore, the description from lines 412 to 414 in the authors' revised version also needs to be revised.

Reviewer #2:

Remarks to the Author:

The authors have considered the issues raised in my previous review, and in my view have addressed my previous concerns satisfactorily.

However, I agree with concerns of other reviewers regarding the localization of E2 in this system. In particular, more convincing data is needed to show that transfected E2 is expressed on the cell surface (e.g. by flow cytometry rather than microscopy-based co-localization). During infection, it is possible that a small fraction of E2 ends up on the cell surface (indeed, some groups have reported this) but the majority of E2 would be intracellular and thus how it activates EGFR needs further consideration.

The data could also be complemented with HCV pseudoparticle data which would better reflect what happens during infection (e.g., E2-EGFR likely interact during the entry process when viral particles are in contact with the cell surface)

Reviewer #3:

Remarks to the Author:

I continue having the following simple issues not cleared by the authors. Please clarify the manuscript.

Comments:

1. The results describe the use of enveloped viruses, but most of them focus upon the use of cell transfection. The HCV E2 full-length protein expressed in these cells would not likely be secreted or expressed on the cell surface. Therefore, the lipid envelope would unlikely be important. Further, the results do not adequately address the split role of the EGFR pathway (Jak-Stat/Akt).
2. The use of multiple viruses is somewhat distracting as their mechanisms of function may differ (i.e. secreted, cell surface vs. ER retained protein) and are not elaborated.
9. P. 14, line 382-386: It remains unclear how the lipid envelope is working here. Is the transfected HCV E2 sequence from ectopic expression of cells transported to cell surface, secreted, or retained in the ER of the cells for having retention signal? If it is retained in cells in its native E2 form what role does the viral envelope play here?

I still have concerns with respect to responses to the above 3 comments directed at a similar issue. The citations (#9 and 10) provided in response from the authors do not suggest that HCV E2 is expressed on cell surface in its unmodified (native) sequence. Additionally, the supportive fluorescence evidence provided by the authors may not reflect surface expression from stained cells as they were fixed before the staining procedure. I would think this set of data reflects intracellular expression of E2. The observed effect may likely reflect manifestation from ER retained E2 protein! And would not make a change in the main observation of the manuscript. The authors should appropriately revise the manuscript if they believe that the E2 protein is expressed in cells.

Responses to the Reviewers

We thank the reviewers for their valuable comments. We have revised the manuscript according to these comments. Please note that all the changes made in the revised manuscript were shown with red color. In the following, please find our point-to-point responses.

Reviewer #1

The authors revised their manuscript very well. Just one point should be amended.

In the authors' response to comment 6, there is a statement that "Fut8KO ICR mice are embryonically lethal and difficult to breed" but the embryonically lethal mice are C57BL/6J mice, not ICR mice. Fujii H et al. conducted research using ICR Fut8KO mice. Please refer to Gastroenterology 2016. Therefore, the description from lines 412 to 414 in the authors' revised version also needs to be revised.

Response: Thanks for the kind reminder. We have changed "Since Fut8KO in ICR background mouse... model in the following experiments." to "We used Fut8 tissue-specific knockdown in HCV-infected ICR^{4R+} transgenic mouse model or the FUT8 inhibitor 2FF in VSV-infected C57BL/6J mouse infection model in the following experiments" shown in red color in the 2nd revision (Lines 402-404).

Reviewer #2

The authors have considered the issues raised in my previous review, and in my view have addressed my previous concerns satisfactorily.

However, I agree with concerns of other reviewers regarding the localization of E2 in this system. In particular, more convincing data is needed to show that transfected E2 is expressed on the cell surface (e.g. by flow cytometry rather than microscopy-based co-localization). During infection, it is possible that a small fraction of E2 ends up on

the cell surface (indeed, some groups have reported this) but the majority of E2 would be intracellular and thus how it activates EGFR needs further consideration.

The data could also be complemented with HCV pseudoparticle data which would better reflect what happens during infection (e.g., E2-EGFR likely interact during the entry process when viral particles are in contact with the cell surface)

Response: Thank you for the helpful suggestion. According to the reviewer's suggestion, we have done new experiments with HCV assessing whether E2-EGFR likely interact during the entry process when viral particles are in contact with the cell surface EGFR. We found that HCV-E2 interacted with EGFR after 15 min upon infection, and both HCV-E2 and EGFR transferred from the cell surface into the cell interior (inducing EGFR internalization) by both confocal microscopy (**Figure S3C**) and flow cytometry analysis (**Figure S3D-S3E**). And as the reviewer suggested, we also found that, at 48 h post infection, the majority of EGFR and a small fraction of E2 ended up on the cell surface, while the majority of E2 was intracellular (**Figure S3C-S3E**). We have presented the new data in the revised manuscript shown in red color (Lines 196-203).

(C) Huh7 cells were infected with HCV (MOI = 10) for the indicated time. Cells were fixed and probed for EGFR and HCV-E2. DAPI was used to stain cellular nuclei. Representative confocal microscopy images are shown. (D-E) FCM analysis images of E2 (D) or EGFR (E) in Huh7 cells infected with HCV (MOI = 10) for indicated time. Representative FCM image of surface and cellular total E2 (D, left panel) or EGFR (E, left panel) stain. Statistical chart of the percentage of E2 (D, right panel) or EGFR (E, right panel) expression were plotted. Isotype control antibodies were used to define background and non-specific binding signal.

Reviewer #3

I continue having the following simple issues not cleared by the authors. Please clarify the manuscript. Comments:

1. The results describe the use of enveloped viruses, but most of them focus upon the use of cell transfection. The HCV E2 full-length protein expressed in these cells would not likely be secreted or expressed on the cell surface. Therefore, the lipid envelope would unlikely be important. Further, the results do not adequately address the split role of the EGFR pathway (Jak-Stat/Akt).

2. The use of multiple viruses is somewhat distracting as their mechanisms of function may differ (i.e. secreted, cell surface vs. ER retained protein) and are not elaborated.

9. P. 14, line 382-386: It remains unclear how the lipid envelope is working here. Is the transfected HCV E2 sequence from ectopic expression of cells transported to cell surface, secreted, or retained in the ER of the cells for having retention signal? If it is retained in cells in its native E2 form what role does the viral envelope play here?

I still have concerns with respect to responses to the above 3 comments directed at a similar issue. The citations (#9 and 10) provided in response from the authors do not

suggest that HCV E2 is expressed on cell surface in its unmodified (native) sequence. Additionally, the supportive fluorescence evidence provided by the authors may not reflect surface expression from stained cells as they were fixed before the staining procedure. I would think this set of data reflects intracellular expression of E2. The observed effect may likely reflect manifestation from ER retained E2 protein! And would not make a change in the main observation of the manuscript. The authors should appropriately revise the manuscript if they believe that the E2 protein is expressed in cells.

Response: Thank you for the helpful suggestion. According to the reviewer's suggestion, we have done new experiments with HCV assessing whether E2-EGFR likely interact during the entry process when viral particles are in contact with the cell surface EGFR. We found that HCV-E2 interacted with EGFR after 15 min upon infection, and both HCV-E2 and EGFR transferred from the cell surface into the cell interior (inducing EGFR internalization) by both confocal microscopy (**Figure S3C**) and flow cytometry analysis (**Figure S3D-S3E**). We also found that, at 48 h post infection, the majority of EGFR and a small fraction of E2 ended up on the cell surface, while the majority of E2 was intracellular (**Figure S3C-S3E**). We have presented the new data in the revised manuscript shown in red color (Lines 196-203).

(C) Huh7 cells were infected with HCV (MOI = 10) for the indicated time. Cells were fixed and probed for EGFR and HCV-E2. DAPI was used to stain cellular nuclei. Representative confocal microscopy images are shown. (D-E) FCM analysis images of E2 (D) or EGFR (E) in Huh7 cells infected with HCV (MOI = 10) for indicated time. Representative FCM image of surface and cellular total E2 (D, left panel) or EGFR (E, left panel) stain. Statistical chart of the percentage of E2 (D, right panel) or EGFR (E, right panel) expression were plotted. Isotype control antibodies were used to define background and non-specific binding signal.

Reviewers' Comments:

Reviewer #2:

Remarks to the Author:

The authors have addressed the previous concerns through addition of flow cytometry data. I am satisfied with the author's revised manuscript.

Reviewer #3:

Remarks to the Author:

The authors satisfactorily responded to the comments and appropriately addressed in the revised manuscript.

Responses to the Reviewers

We thank the reviewers for their valuable comments. In the following, please find our point-to-point responses.

Reviewer #2

The authors have addressed the previous concerns through addition of flow cytometry data. I am satisfied with the author's revised manuscript.

Response: Thank you for the kind comments.

Reviewer #3

The authors satisfactorily responded to the comments and appropriately addressed in the revised manuscript

Response: Thank you for the kind comments.